

# Bootstrapping nonequilibrium stochastic processes

**Minjae Cho**

Leinweber Institute for Theoretical Physics, University of Chicago,
Chicago, Illinois 60637, USA

cho7@uchicago.edu

## Abstract

We show that bootstrap methods based on the positivity of probability measures provide a systematic framework for studying both synchronous and asynchronous nonequilibrium stochastic processes on infinite lattices. First, we formulate linear programming problems that use positivity and invariance property of invariant measures to derive rigorous bounds on their expectation values. Second, for time evolution in asynchronous processes, we exploit the master equation along with positivity and initial conditions to construct linear and semidefinite programming problems that yield bounds on expectation values at both short and late times. We illustrate both approaches using two canonical examples: the contact process in 1+1 and 2+1 dimensions, and the Domany-Kinzel model in both synchronous and asynchronous forms in 1+1 dimensions. Our bounds on invariant measures yield rigorous lower bounds on critical rates, while those on time evolutions provide two-sided bounds on the half-life of the infection density and the temporal correlation length in the subcritical phase.

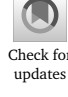

# 1 Introduction

Randomness, whether intrinsic or arising from incomplete information, appears ubiquitously across problems in physics. This naturally leads to the study of the probability distribution (or measure) of states, which provides the notion of physical observables averaged over the randomness. In stochastic processes where time evolution is probabilistic, probability measures that remain invariant under the evolution govern the system's late-time behavior. Such invariant measures are said to be in global balance, where the net probability flux into and out of each state vanishes.

    If an invariant measure further satisfies detailed balance—where the probability flux between any pair of states is balanced—it is called a reversible measure. Stochastic processes

that admit nontrivial reversible measures[1] are referred to as equilibrium, some of which coincide with thermal equilibrium described by the standard Gibbs measure in statistical physics, where the presence of a Hamiltonian aids significantly in understanding the system. Questions such as how a system equilibrates, and what the equilibrium properties are, have led to the discovery of essential concepts and observables, with universality and phase transition being representative examples.

Most physical processes in nature, however, are described by nonequilibrium physics, where nontrivial reversible measures and Hermitian time-evolution generators are absent. In such systems, the focus shifts to invariant measures, and beautiful phenomena like universality and phase transitions still emerge,[2] as demonstrated by many numerical studies. The resulting universality classes are fundamentally distinct from those in equilibrium systems and play a crucial role in our understanding of nature.

From a theoretical standpoint, many elegant tools used to study equilibrium physics—such as reflection positivity—are no longer available. Consequently, our theoretical understanding of nonequilibrium systems still lags behind that of equilibrium systems. For instance, the simple critical nonequilibrium system of directed percolation (DP) in 1+1 dimensions remains unsolved, in stark contrast to the exactly solvable Gibbs measures in one- and two-dimensional Ising models.

Nevertheless, there exist rigorous mathematical results for certain nonequilibrium models, such as the contact process [2] and oriented percolation [3], where properties like monotonicity and duality have proven to be powerful.[3] For example, in the contact process, the existence [2] and nature of the phase transition [5] have been established, and several rigorous methods have been developed to bound physical quantities [2, 4–8]. This motivates the search for a general framework that does not rely on model-specific properties and applies to a broader class of nonequilibrium systems—or even to both equilibrium and nonequilibrium cases.

In this work, we show that the positivity of probability measures under time evolution—a trivial but fundamental property of any stochastic process—provides a natural and systematic method for deriving rigorous bounds on expectation values in nonequilibrium stochastic processes on an infinite lattice. Although positivity alone is trivial, its combination with constraints such as invariance, time evolution, and initial conditions yields nontrivial, rigorous, and sometimes sharp bounds on the expectation values. In the case of invariant measures in equilibrium lattice systems, this idea was recently formulated in [9]. We extend it here to nonequilibrium systems and to noninvariant (time-dependent) measures. The latter generalization is inspired by a recently developed framework for bounding time evolution governed by master equations in stochastic and quantum systems [10–13].[4]

The idea of using basic properties of probability measures (or density matrices in quantum settings), such as positivity, to derive constraints on observables has appeared in various fields of mathematical sciences. Examples include quantum chemistry [15, 16], Markov chains [17], optimal control theory [18], stochastic and dynamical systems [19–23], matrix models [24–29], and classical or quantum lattice systems [9, 30–36]. More broadly, methods based on consistency conditions are referred to as bootstrap methods in the physics literature, with conformal bootstrap [37–39] being a prominent example. This work develops a bootstrap framework for nonequilibrium stochastic processes on infinite lattices, based on the positivity of probability measures.

---

[1] A reversible measure is trivial if the probability flux between all pairs of states is zero. An example of such a trivial reversible measure is the absorbing state, which will be discussed later in this work.

[2] See e.g. [1] and references therein for a review of the subject.

[3] See Chapters II, III, and VI of [4] for an introduction to some of these mathematical tools and results.

[4] See [14] and references therein for a more complete overview of the subject.

## 1.1 Setup

We begin by introducing the class of problems studied in this work. We consider the stochastic time evolution of spin configurations on an infinite lattice $\mathbb{Z}^d$. Concretely, each site $i \in \mathbb{Z}^d$ carries a spin degree of freedom $s_i \in \{1, -1\}$. A spin configuration, or a state, $s$ is then an element of the space $S = \{1, -1\}^{\mathbb{Z}^d}$. As time evolves, a state $s \in S$ undergoes probabilistic transitions to other states in $S$. We distinguish two types of time evolution: asynchronous and synchronous.

### 1.1.1 Asynchronous stochastic processes

In asynchronous processes, time is continuous, $t \in \mathbb{R}$, and transitions from $s$ to $s' \in S$ occur spontaneously. These processes are also referred to as *interacting particle systems* [4]. We focus on processes where $s$ may transition to $s'$ only if they differ at a single site of $\mathbb{Z}^d$. As a result, each state $s$ has only countably many states it can transition to.

Let $\bar{s}^i$ denote the state that differs from $s$ only at site $i$, that is, $\bar{s}^i_j = (1 - 2\delta_{ij})s_j$ for all $j \in \mathbb{Z}^d$. We assume the transition rate $c(i, s) \geq 0$ from $s$ to $\bar{s}^i$ depends only on $s$ and the site $i$ being updated. Furthermore, we consider only local processes, meaning that $c(i, s)$ depends on finitely many spins near site $i$. Specific choices of $c(i, s)$ define different stochastic processes.

If the system is instead defined on a finite subset $\Lambda \subset \mathbb{Z}^d$, so that the state space is finite, the time evolution of the probability measure $\Pi(s_\Lambda)$ over $s_\Lambda \in \{1, -1\}^\Lambda$ is governed by the master equation (also called the Kolmogorov equation):

$$\frac{d}{dt}\Pi(s_\Lambda) = \sum_{i \in \Lambda} c(i, \bar{s}^i_\Lambda)\Pi(\bar{s}^i_\Lambda) - \sum_{i \in \Lambda} c(i, s_\Lambda)\Pi(s_\Lambda), \tag{1}$$

where the first and second terms on the right-hand side (RHS) represent the gain and loss of probability weight on the state $s_\Lambda$, respectively. The expectation value $\langle f(s_\Lambda) \rangle = \sum_{s_\Lambda \in \{1,-1\}^\Lambda} \Pi(s_\Lambda) f(s_\Lambda)$ of any function $f(s_\Lambda)$ then satisfies

$$\begin{aligned} \frac{d}{dt}\langle f(s_\Lambda) \rangle &= \sum_{s_\Lambda \in \{1,-1\}^\Lambda} \left( \sum_{i \in \Lambda} c(i, \bar{s}^i_\Lambda)\Pi(\bar{s}^i_\Lambda)f(s_\Lambda) - \sum_{i \in \Lambda} c(i, s_\Lambda)\Pi(s_\Lambda)f(s_\Lambda) \right) \\ &= \sum_{i \in \Lambda} \left\langle c(i, s_\Lambda)\left( f(\bar{s}^i_\Lambda) - f(s_\Lambda) \right) \right\rangle. \end{aligned} \tag{2}$$

While the infinite-lattice version of equation (1) is not well-defined, the expectation value form (2) does admit an extension:

$$\text{Master equation:} \quad \frac{d}{dt}\langle f(s) \rangle = \sum_{i \in \mathbb{Z}^d} \left\langle c(i, s)\left( f(\bar{s}^i) - f(s) \right) \right\rangle. \tag{3}$$

When $f(s)$ depends on only finitely many spins, the RHS sum truncates to a finite number of terms. The master equation (3) describes how expectation values evolve in time, given initial conditions.

For an invariant measure, the expectation values are time-independent and satisfy

$$\text{Invariance equation:} \quad \sum_{i \in \mathbb{Z}^d} \left\langle c(i, s)\left( f(\bar{s}^i) - f(s) \right) \right\rangle = 0, \quad \forall f(s) \in D(S), \tag{4}$$

where $D(S)$ is a function space called the core; see Chapter I of [4] for a formal definition. For our purposes, it suffices to know that the set of finite-degree polynomials in $\{s_i\}_{i \in \mathbb{Z}^d}$ is dense in $D(S)$. If equation (4) is satisfied term by term—i.e. $\left\langle c(i, s)\left( f(\bar{s}^i) - f(s) \right) \right\rangle = 0$ for

each $i \in \mathbb{Z}^d$—then the measure is called reversible. A stochastic process is called equilibrium if it admits nontrivial reversible measures; otherwise, it is nonequilibrium. We now introduce two examples of asynchronous, nonequilibrium stochastic processes that will serve as the main examples of this work.

### 1.1.2 Contact process

The first example is the contact process on $\mathbb{Z}^d$, which models the spread of an epidemic. A spin at site $i$ is considered "healthy" if $s_i = -1$ and "infected" if $s_i = 1$.[5] The transition from $s$ to $\bar{s}^i$ follows a simple rule: 1) if site $i$ is infected, it becomes healthy at rate 1; and 2) if site $i$ is healthy, it becomes infected at rate $\lambda n_i$, where $n_i$ is the number of infected nearest neighbors. The infection rate $\lambda \in \mathbb{R}_+$ is a parameter of the process.

The corresponding transition rate is given by

$$c(i,s) = \frac{1+s_i}{2} + \lambda \frac{1-s_i}{2} \sum_{j \in N(i)} \frac{1+s_j}{2}, \tag{5}$$

where $N(i) = \{j \in \mathbb{Z}^d \mid \|j - i\|_2 = 1\}$ denotes the set of nearest neighbors of site $i$. The first term on the RHS is nonzero only when $s_i = 1$ and represents the rate-1 recovery to $s_i = -1$. The second term is active only when $s_i = -1$ and describes the infection transition. The sum $\sum_{j \in N(i)} \frac{1+s_j}{2}$ counts the number of infected neighbors, so that the total infection rate is $\lambda n_i$ as expected.

There is a special state called the "absorbing state" where $s_i = -1$ for all $i \in \mathbb{Z}^d$—i.e., all sites are healthy. Regardless of the value of $\lambda$, once the system enters the absorbing state, it remains there permanently. The absorbing state defines a trivial invariant (and reversible) measure. The central question is whether there exist other, nontrivial invariant measures.[6]

It was shown in [2] that there exists a finite critical infection rate $\lambda_c$ such that for all $\lambda < \lambda_c$ (the subcritical phase), the absorbing state is the unique invariant measure; whereas for $\lambda > \lambda_c$ (the supercritical phase), there also exists a nontrivial invariant measure known as the upper invariant measure. For example, in $d = 1$, numerical studies suggest $\lambda_c \approx 1.6491$ [40], with rigorous bounds $1.5388 \le \lambda_c$ [6] and $\lambda_c \le 1.942$ [7]. The transition at $\lambda = \lambda_c$ has been proven to be continuous [5], and is believed to lie in the DP universality class in 1+1 dimensions. These rigorous results often rely on specific properties of the contact process, such as monotonicity and duality.[7]

Because the contact process (and the Domany-Kinzel model discussed next) is translation-invariant, we restrict our attention to translation-invariant measures in this work. The infection density is then defined by

$$\rho = \left\langle \frac{1+s_i}{2} \right\rangle. \tag{6}$$

For $\lambda \le \lambda_c$, $\rho$ decays to zero over time regardless of the initial condition. In contrast, for $\lambda > \lambda_c$, the upper invariant measure exhibits $\rho > 0$, so the infection density does not necessarily vanish over time. Thus, $\rho$ serves as the order parameter for the phase transition.

### 1.1.3 Asynchronous Domany-Kinzel model

The second example we study in this work is the asynchronous Domany-Kinzel model on $\mathbb{Z}$. The original version of the model [41, 42] is synchronous, which we will introduce shortly. As

---

[5]This differs from the usual convention in the literature, where the healthy state is assigned value 0. The map between $s_i \in \{1, -1\}$ used in this work and $\eta_i \in \{0, 1\}$ used elsewhere is simply $\eta_i = \frac{1+s_i}{2}$.

[6]Since the invariance equations (4) are linear in the measure, any convex combination of invariant measures is also invariant, implying the existence of infinitely many invariant measures if more than one exists.

[7]See e.g. Chapters II, III, and VI of [4].

before, we adopt the terminology of sites being either healthy or infected. The transition rule at site $i$ is as follows: 1) if exactly one of its nearest neighbors is infected, then site $i$ becomes infected at rate $p_1$ and becomes healthy at rate $1-p_1$; 2) if both nearest neighbors are infected, then site $i$ becomes infected at rate $p_2$ and healthy at rate $1-p_2$; 3) if none of the neighbors are infected, then site $i$ becomes healthy at rate 1. Here, both $p_1$ and $p_2$ are real parameters taking values in $[0,1]$. The explicit expression for the transition rate is

$$
\begin{aligned}
c(i,s) = {} & \frac{1-s_i}{2}\left(p_1\frac{1-s_{i-1}s_{i+1}}{2} + p_2\frac{1+s_{i-1}}{2}\frac{1+s_{i+1}}{2}\right) \\
& + \frac{1+s_i}{2}\left((1-p_1)\frac{1-s_{i-1}s_{i+1}}{2} + (1-p_2)\frac{1+s_{i-1}}{2}\frac{1+s_{i+1}}{2} + \frac{1-s_{i-1}}{2}\frac{1-s_{i+1}}{2}\right).
\end{aligned}
\tag{7}
$$

As in the contact process, the model has a unique absorbing state for all values of $p_1$ and $p_2$, in which every site is healthy. The synchronous version of the Domany-Kinzel model is well known to exhibit a curve of DP criticality in the $(p_1, p_2)$ parameter space (see e.g. [1]). By the Janssen-Grassberger conjecture [43, 44], the asynchronous version is also expected to belong to the DP universality class. For any fixed $p_2 \in [0,1)$, there exists a critical value $p_{1c}$ such that for $p_1 < p_{1c}$, the absorbing state is the only invariant measure, while for $p_1 > p_{1c}$, a nontrivial upper invariant measure appears. As in the contact process, the infection density $\rho$ serves as the order parameter.

The model exhibits two distinct regimes: 1) the monotonic regime, $p_1 \le p_2$, and 2) the non-monotonic regime, $p_1 > p_2$. In the non-monotonic case, having both neighbors infected leads to a *smaller* infection rate than having only one infected neighbor.[8] This behavior invalidates many standard theoretical tools used for monotonic processes such as the contact process.[9] To illustrate the power of the bootstrap method developed in this work, we focus on the non-monotonic case with $p_2 = 0$.

### 1.1.4 Synchronous stochastic processes

In synchronous processes, time is discrete, $t \in \mathbb{N}$, and all spins in the configuration are updated simultaneously. We restrict attention to local Markov processes, in which the spin at site $i$ at time $t+1$ depends only on the spins at nearby sites (e.g., nearest neighbors) at time $t$. Such processes are also known as *probabilistic cellular automata*.

Rather than discussing the general case, we directly introduce the synchronous Domany-Kinzel model on $\mathbb{Z}$. In this model, the time evolution of spin configurations $s(t)$ is governed by the following update rule:[10]

$$
\text{For all } i \in \mathbb{Z} \text{ and } t \in \mathbb{N}, \quad s_i(t+1) = \begin{cases} 1, & \text{with probability } p_1 \text{ if } s_i(t) + s_{i+1}(t) = 0, \\ 1, & \text{with probability } p_2 \text{ if } s_i(t) + s_{i+1}(t) = 2, \\ -1, & \text{otherwise.} \end{cases}
\tag{8}
$$

Special cases include: $p_1 = p_2$ (site percolation), $p_2 = p_1(2-p_1)$ (bond percolation), and $p_2 = 0$ (Wolfram's rule 18 [46]). As in the asynchronous examples discussed earlier, the synchronous Domany-Kinzel model also possesses a unique absorbing state with $s_i(t) = -1$ for all $i \in \mathbb{Z}$, and is expected to exhibit a line of DP criticality in the $(p_1, p_2)$ parameter space separating subcritical and supercritical phases.

---

[8]For a more complete definition of monotonicity, we refer the reader to Chapters II and III of [4].

[9]Nonetheless, there are established mathematical results for certain non-monotonic processes; see, e.g., [45].

[10]This definition of the Domany-Kinzel model is equivalent to the traditional one in which $s_i(t+1)$ depends only on $s_{i-1}(t)$ and $s_{i+1}(t)$, since the odd and even sublattices in $\mathbb{Z}$ at a given time $t$ are dynamically decoupled in that case.

Define $D_L = \{1, 2, \ldots, L\} \subset \mathbb{Z}$ and let $P_L$ denote the set of polynomials in the spins $\{s_i\}_{i \in D_L}$. For any $f(s) \in P_L$, the update rule (8) implies that the expectation value evolves according to

$$\langle f(s(t+1)) \rangle = \sum_{A \subset D_{L+1}} C_A(f) \left\langle \prod_{i \in A} s_i(t) \right\rangle, \tag{9}$$

where the sum is over all subsets $A$ of $D_{L+1}$, and the coefficients $C_A(f) \in \mathbb{R}$ depend on $A$ and on $f(s)$, as determined by the update rule (8).

As a concrete example, the probability that $s_i(t+1) = 1$ equals the sum of $p_1 \times$ (probability that exactly one of $s_i(t)$ and $s_{i+1}(t)$ is 1), and $p_2 \times$ (probability that both $s_i(t)$ and $s_{i+1}(t)$ are 1), which gives

$$\left\langle \frac{1 + s_i(t+1)}{2} \right\rangle = p_1 \left\langle \frac{1 - s_i(t)s_{i+1}(t)}{2} \right\rangle + p_2 \left\langle \frac{1 + s_i(t)}{2} \frac{1 + s_{i+1}(t)}{2} \right\rangle. \tag{10}$$

By similarly expressing probabilities for all spin configurations on $D_L$, one can compute all the coefficients $C_A(f)$ in (9) for each $f(s) \in P_L$. We will discuss the relationship between these expectation values and probabilities in more detail shortly.

Invariant measures satisfy the condition

$$\text{Invariance equation:} \quad \langle f(s) \rangle = \sum_{A \subset D_{L+1}} C_A(f) \left\langle \prod_{i \in A} s_i \right\rangle, \quad \forall f(s) \in P_L, \quad \forall L \in \mathbb{N}. \tag{11}$$

In this work, we restrict to translation-invariant measures, in which case equation (11) serves as the definition of invariance. As with the asynchronous cases, the absorbing state is the unique invariant measure in the subcritical phase, while a nontrivial upper invariant measure appears in the supercritical phase. The model is monotonic for $p_1 \leq p_2$ and non-monotonic for $p_1 > p_2$. In this work, we focus on the non-monotonic case $p_2 = 0$ corresponding to Wolfram's rule 18.

## 1.2 Main ideas of the bootstrap method

We now briefly outline the bootstrap method for the nonequilibrium stochastic processes discussed above. The key observation is that the invariance equations (4) and (11), as well as the master equation (3), are all *linear* in the expectation values. The only distinction between these expectation values and arbitrary real numbers or time-dependent functions that satisfy the same linear equations is that the former arise from a valid probability measure. This distinction is enforced by the following positivity constraints, which we refer to as probability bounds:

$$\text{Probability bound:} \quad \left\langle \prod_{i \in A} \frac{1 + u_i s_i}{2} \right\rangle \geq 0, \tag{12}$$

for any finite subset $A \subset \mathbb{Z}^d$ and any spin assignment $u \in \{1, -1\}^A$. The function $\prod_{i \in A} \frac{1 + u_i s_i}{2}$ is the indicator function for the event $\{s \in S \mid s_i = u_i, \ \forall i \in A\}$; that is, it equals 1 if the event occurs and 0 otherwise. Therefore, its expectation value corresponds to the probability of that event and must be nonnegative.

We have already encountered such indicator functions in the expressions for the transition rates (5) and (7), the definition of the infection density $\rho$ in (6), and the time evolution of probabilities in the synchronous case (10). Crucially, the probability bounds are themselves *linear* in the expectation values.

### 1.2.1 Invariant measures

Recently in [9], equations (4) and (12) were combined to formulate the following linear programming (LP) problem for determining invariant measures of asynchronous stochastic processes:

$$
\text{Over the space of expectation values } \left\langle \prod_{i \in A} s_i \right\rangle \text{ for finite subsets } A \text{ of } \mathbb{Z}^d,
$$

minimize $\langle q(s) \rangle$, where $q(s)$ is a polynomial of interest, subject to

linearity of expectation values, normalization $\langle 1 \rangle = 1$, (4), and (12).
$$\tag{13}$$

Additional symmetry constraints, such as translation invariance, may be included as they are also linear in the expectation values. Although (13) is, in principle, an optimization problem with infinitely many variables and constraints, we may focus on a finite subset of variables and constraints that still *must* be satisfied. The resulting minimum $\langle q(s) \rangle_{min}$ from such a finite relaxation then provides a *rigorous* lower bound on the exact value of $\langle q(s) \rangle$ for *any* invariant measure on $\mathbb{Z}^d$. A rigorous upper bound can be obtained analogously by maximizing the same objective.

In [9], it was shown that as more variables and constraints are systematically included, the solutions of (13) converge to the expectation values realized by an actual invariant measure. This convergence theorem implies that (13) may serve as an alternative definition of invariant measures.

An analogous LP problem applies to invariant measures of synchronous processes governed by the update rule (8):

$$
\text{Over the space of expectation values } \left\langle \prod_{i \in A} s_i \right\rangle \text{ for finite subsets } A \text{ of } \mathbb{Z},
$$

minimize $\langle q(s) \rangle$, where $q(s)$ is a polynomial of interest, subject to

linearity, normalization, translation invariance, (11), and (12).
$$\tag{14}$$

As in the asynchronous case, this LP yields a rigorous lower bound on $\langle q(s) \rangle$ for all invariant measures. The convergence theorem from [9] extends straightforwardly to this setting as well.

For all stochastic processes considered in this work, every invariant measure in the supercritical phase is a linear combination of the absorbing state and the upper invariant measure. Consequently, its expectation value $\langle \cdots \rangle^w$ is given by a weighted average of $\langle \cdots \rangle^{abs}$ and $\langle \cdots \rangle^{up}$, which are the expectation values corresponding to the absorbing state and the upper invariant measure, respectively:

$$
\left\langle \prod_{i \in A} s_i \right\rangle^{w} = (1-w) \left\langle \prod_{i \in A} s_i \right\rangle^{abs} + w \left\langle \prod_{i \in A} s_i \right\rangle^{up}
$$
$$
= (1-w) \, (-1)^{|A|} + w \left\langle \prod_{i \in A} s_i \right\rangle^{up}, \qquad \forall A \subset \mathbb{Z}^d, \tag{15}
$$

where $|A|$ denotes the number of sites in $A$, and the weight $w$ lies in $[0, 1]$. In the analysis below, we focus on the case $w > 0$.

To derive results that apply specifically to the upper invariant measure,[11] we rewrite

$$
\left\langle \prod_{i \in A} s_i \right\rangle^{w} = \left\langle \prod_{i \in A} s_i \right\rangle^{abs} + \mathcal{D}\left( \prod_{i \in A} s_i \right), \quad \mathcal{D}\left( \prod_{i \in A} s_i \right) = w \left( \left\langle \prod_{i \in A} s_i \right\rangle^{up} - \left\langle \prod_{i \in A} s_i \right\rangle^{abs} \right). \tag{16}
$$

---

[11]We thank Yuan Xin for suggestions and extended discussions on the bootstrap analysis of the upper invariant measure.

Up to a factor of $w$, the quantity $\mathcal{D}$ represents the difference between the expectation values for the upper invariant measure and the absorbing state. Since the absorbing state is invariant, the invariance equations (4) and (11) for $\langle \cdots \rangle^w$ are homogeneous and linear in $\mathcal{D}$ (note that $\mathcal{D}(1) = 0$).

Next, consider probabilities of events where at least one spin takes the value 1. These probabilities are identically zero in the absorbing state. Therefore, when $w > 0$, their positivity imposes nontrivial constraints on the upper invariant measure alone, which are homogeneous and linear in $\mathcal{D}$:

$$\left\langle \prod_{i \in A} \frac{1 + u_i s_i}{2} \right\rangle^w = \mathcal{D}\left( \prod_{i \in A} \frac{1 + u_i s_i}{2} \right) \geq 0, \quad \forall A \subset \mathbb{Z}^d \text{ s.t. } \exists i \in A, \quad \text{where } u_i = 1. \tag{17}$$

Since both the invariance equations and the probability bounds in (17) are homogeneous and linear in $\mathcal{D}$, they induce nontrivial linear constraints on the ratios. For convenience, we consider the ratio of $\mathcal{D}$ acting on a generic polynomial to $\mathcal{D}(s_i)$; that is, $\mathcal{D}(s_i)$ serves as a reference point:

$$\mathcal{R}\left( \prod_{i \in A} s_i \right) = \frac{\mathcal{D}\left( \prod_{i \in A} s_i \right)}{\mathcal{D}(s_i)} = \frac{\left\langle \prod_{i \in A} s_i \right\rangle^{up} - (-1)^{|A|}}{\langle s_i \rangle^{up} + 1}, \quad \forall A \subset \mathbb{Z}^d, \tag{18}$$

where we have assumed translation invariance in writing $\langle s_i \rangle^{up}$. Since the denominator $\langle s_i \rangle^{up} + 1$ is strictly positive, the probability bounds (17) remain valid even when $\mathcal{D}$ is replaced by $\mathcal{R}$:

$$\mathcal{R}\left( \prod_{i \in A} \frac{1 + u_i s_i}{2} \right) \geq 0, \quad \forall A \subset \mathbb{Z}^d \text{ s.t. } \exists i \in A, \quad \text{where } u_i = 1. \tag{19}$$

Similarly, the invariance equations (4) and (11) can be expressed purely in terms of $\mathcal{R}$, and remain linear in $\mathcal{R}$.

The LP problem that yields nontrivial bounds on $\mathcal{R}$, and hence on the upper invariant measure, is given by:

> Over the space of ratios $\mathcal{R}\left( \prod_{i \in A} s_i \right)$ for finite subsets $A$ of $\mathbb{Z}$,
>
> minimize $\mathcal{R}(q(s))$, where $q(s)$ is a polynomial of interest, subject to (20)
>
> linearity, $\mathcal{R}(1) = 0$, $\mathcal{R}(s_i) = 1$, translation invariance, (19),
>
> and invariance (4) or (11) with (16) and (18).

In practice, convex optimization solvers for large-scale problems typically return results with rounding errors. However, due to the simplicity of LP formulations like (13), (14), and (20), one can employ simplex methods to obtain *exact* solutions. In this work, we use the built-in `LinearOptimization` function in *Mathematica* [47], which provides an exact LP solver for problems such as (13), (14), and (20). The resulting bounds are mathematically rigorous bounds on the expectation values and their ratios $\mathcal{R}$.

### 1.2.2 Noninvariant measures: Short-time behavior

We now consider noninvariant measures for asynchronous processes whose expectation values evolve over time $t$. To make this time dependence explicit, we denote them as $\langle \cdots \rangle_t$. The linear constraints (3) and (12) must hold at all times $t \in \mathbb{R}$.

Suppose we specify initial conditions $\langle \prod_{i \in A} s_A \rangle_{t=0} = v_A$, which are clearly *linear* in the expectation values. This leads to the following convex optimization problem:

Over the space of $\left\langle \prod_{i \in A} s_i \right\rangle_t$ for finite subsets $A$ of $\mathbb{Z}^d$ and $t \in [0, T]$ for fixed $T > 0$,

minimize $\langle q(s) \rangle_{t=T}$, where $q(s)$ is a polynomial of interest, subject to

linearity of expectation values, $\langle 1 \rangle_t = 1$, (3) and (12) for all $t \in [0, T]$, $\qquad$ (21)

and initial conditions $\left\langle \prod_{i \in A} s_A \right\rangle_{t=0} = v_A$.

Symmetry constraints respected by both the time evolution and the initial conditions may also be included, as they are linear in the expectation values. The variables in this optimization problem are once-differentiable *functions* of $t \in [0, T]$, so even when restricted to spins over finite subsets $A \subset \mathbb{Z}^d$, the problem remains infinite-dimensional.

In section 3.2, we consider the dual convex optimization problem, which can be made finite-dimensional while still yielding rigorous lower bounds on $\langle q(s) \rangle_{t=T}$. Analogously, an upper bound is obtained by maximizing the primal objective. The rigorous nature of these bounds is guaranteed by the standard weak duality theorem in convex optimization. This approach has recently been used in [12–14] to derive similar bounds for stochastic and quantum systems.

Depending on the specific formulation, the dual problem can take the form of either LP or semidefinite programming (SDP). The former yields mathematically rigorous bounds, while the latter offers bounds up to rounding errors but with significantly reduced computation time. In this work, we employ the built-in `SemidefiniteOptimization` function in *Mathematica* [47], using `Method → "MOSEK"` [48] for solving the SDP.

### 1.2.3 Noninvariant measures: Late-time behavior

In the subcritical phase, expectation values decay exponentially fast to those of the absorbing state at late times, for any initial conditions with $\langle s_i \rangle_{t=0} > -1$:

$$\text{As } t \to \infty, \quad \left\langle \prod_{i \in A} s_i \right\rangle_t \to (-1)^{|A|} + B_A e^{-\frac{t}{\xi}}, \quad \forall A \subset \mathbb{Z}^d, \qquad (22)$$

for some real numbers $B_A$ satisfying $(-1)^{|A|} B_A < 0$. Here, $\xi > 0$ is the temporal correlation length, whose inverse $\Delta = \xi^{-1}$ is the spectral gap of the time-evolution generator. Crucially, the decay exponent $\xi$ is the same for all expectation values.

The master equation (3) and probability bounds (12) must hold even at very late times. Therefore, we can substitute (22) into them, where $\xi$ appears through the $\frac{d}{dt} \langle f(s) \rangle$ term in the master equation (3). These provide constraints that $\xi$ must satisfy. At a trial value of $\xi$, the problem of determining whether these constraints are satisfied reduces to a simple LP problem:

Over the space of $B_A$ for finite subsets $A$ of $\mathbb{Z}^d$ at a trial value $\xi > 0$,

maximize $B_{\{i\}}$ subject to

symmetries, linearity of expectation values, $\langle 1 \rangle_t = 1$, $B_{\{i\}} \leq 1$, $\qquad$ (23)

and (3) and (12) with the substitution (22) with $t \to \infty$.

Note that we have added the constraint $B_{\{i\}} \leq 1$. Without it, the constraints are homogeneous (and in fact linear) in $B_A$, so the problem would be unbounded unless the solution is $B_A = 0$. Therefore, there are only two possible outcomes of the LP (23): either $B_{\{i\}} = 0$ or $B_{\{i\}} = 1$. If

the result is $B_{\{i\}} = 0$, which contradicts $(-1)^{|A|}B_A < 0$, then the trial value of $\xi$ is excluded as a possible temporal correlation length. In contrast, if the result is $B_{\{i\}} = 1$ and the inequality $(-1)^{|A|}B_A < 0$ can be explicitly verified, then the trial value of $\xi$ is allowed.

## 1.3 Sample results and outline

We briefly present a few sample results of the bootstrap method here, deferring more complete sets of results to the main sections. Applying the LP (13) for invariant measures to the contact process on $\mathbb{Z}^2$ at $\lambda = 1$, we obtain the upper bound on the infection density

$$\rho \leq \frac{1915290}{2610007} \approx 0.733826\,, \tag{24}$$

which is consistent with the kinetic Monte Carlo (KMC) estimate $\rho \approx 0.72506(26)$ discussed in appendix B. For certain values of $\lambda$, the upper bound on $\rho$ becomes 0, implying that $\rho = 0$ and thereby providing a lower bound on the critical value $\lambda_{c,2}$ for the phase transition on $\mathbb{Z}^2$. This yields

$$\lambda_{c,2} \geq 0.362\,, \tag{25}$$

which is consistent with the Monte Carlo estimate $\lambda_{c,2} \approx 0.41220(3)$ [49].

We similarly apply the LP (14) for invariant measures of the synchronous Domany-Kinzel model on $\mathbb{Z}$ at $p_2 = 0$ (Wolfram's rule 18), and obtain the following upper bound on $\rho$ at $p_1 = 0.9$:

$$\rho \leq 0.454362\,, \tag{26}$$

which is consistent with the Monte Carlo estimate $\rho \approx 0.42621(33)$. We also obtain a lower bound on the critical value $p_{1c}^s$ for $p_1$ given by

$$p_{1c}^s \geq 0.772\,, \tag{27}$$

consistent with the estimate $p_{1c}^s \approx 0.799(2)$ from [50].

To obtain nontrivial bounds on the upper invariant measure in the supercritical phase, we apply the LP (20) to the contact process on $\mathbb{Z}$ at $\lambda = 2$ and find

$$-0.6404661 < \mathcal{R}(s_1 s_3) = \frac{\langle s_1 s_3 \rangle^{up} - 1}{\langle s_1 \rangle^{up} + 1} < -0.6403856\,, \tag{28}$$

which is consistent with the KMC estimate $\mathcal{R}(s_1 s_3) \approx -0.6403(7)$.

Turning to time evolution, we apply (21) to the contact process on $\mathbb{Z}$ at $\lambda = 2$, and obtain the following bounds from the dual SDP problems, where the initial conditions are $\langle s_A \rangle_{t=0} = 0$ for all $A \subset \mathbb{Z}$:

$$0.61617 \leq \rho_{t=1} \leq 0.61880\,. \tag{29}$$

Here, $\rho_t$ denotes the infection density at time $t$.

For the contact process, $\rho_t$ is a non-increasing function of time $t$ when the initial condition is $\left\langle \prod_{i \in A} s_i \right\rangle_{t=0} = 1$ for all $A \subset \mathbb{Z}^d$.[12] In the subcritical phase, we define the half-life $t_{1/2}$ for such initial conditions by $\rho_{t=t_{1/2}} = \frac{1}{2}$. If, at a given $t = T_1$, the lower bound on $\rho_{t=T_1}$ is greater than $\frac{1}{2}$, then $T_1 \leq t_{1/2}$. Similarly, if the upper bound on $\rho_{t=T_2}$ is less than $\frac{1}{2}$, then $t_{1/2} \leq T_2$. This yields the following two-sided bounds on $t_{1/2}$ for the contact process on $\mathbb{Z}$ at $\lambda = 1$:

$$1.575 < t_{1/2} < 1.59\,, \tag{30}$$

consistent with the KMC estimate $t_{1/2} \approx 1.589(14)$.

---

[12]See Theorem 2.3 in chapter III of [4].

Lastly, we apply (23) to derive bounds on the temporal correlation length $\xi$. For example, in the contact process on $\mathbb{Z}^2$ at $\lambda = 0.1$, we obtain

$$1.48 < \xi < 1.528 \,, \tag{31}$$

while a rough KMC estimate gives $\xi \approx 1.411$.

This paper is organized as follows. In section 2, we introduce the LP formulation for invariant measures and derive rigorous bounds on their expectation values, which further lead to lower bounds on the critical rates. In section 3, we discuss the convex optimization problem for the short-time evolution of the expectation values, whose dual problem provides bounds on them. These bounds then yield two-sided bounds on the half-life. We next present the LP formulation for the late-time evolution in section 4, which provides two-sided bounds on the temporal correlation length. We conclude with future prospects in section 5.

## 2 Bootstrapping the invariant measures

We start by describing hierarchies of LP problems that impose the defining properties of the invariant measures of the stochastic processes of interest, along with the resulting bounds on their expectation values.

### 2.1 LP hierarchy for asynchronous stochastic processes on the lattice $\mathbb{Z}$

For asynchronous stochastic processes, the LP hierarchy for the invariant measures was constructed in [9], which we now review. We begin with systems on $\mathbb{Z}$ and discuss the $\mathbb{Z}^2$ case in section 2.3. Recall the notations $D_L = \{1, \ldots, L\} \subset \mathbb{Z}$ and $P_L$, the set of polynomials of spins $\{s_i\}_{i \in D_L}$.

Both the contact process and the asynchronous Domany-Kinzel model on $\mathbb{Z}$ respect three types of symmetries on the lattice: 1. translation, 2. reflection about a lattice site, and 3. reflection about a midpoint between two lattice sites. More concretely, given a finite subset $A \subset \mathbb{Z}$, right- and left-translations $\tau_+$ and $\tau_-$ act as $\tau_\pm(A) = \{i \pm 1 | i \in A\}$; reflection about a lattice site $j$, denoted $r_j$, acts as $r_j(A) = \{j - i | i \in A\}$; and reflection about a midpoint between sites $j$ and $j+1$, denoted $v_j$, acts as $v_j(A) = \{j - i + 1 | i \in A\}$. We define the equivalence relation $\sim$ between two finite subsets $A$ and $B$ of $\mathbb{Z}$ as: $A \sim B$ if and only if $A$ and $B$ can be obtained from each other via repeated actions of $\tau_\pm$, $r_j$, and $v_j$ for $j \in \mathbb{Z}$.

The LP hierarchy $LP_{inv}$ for the invariant measures, respecting the symmetries of the lattice, for the transition rate $c(i, s)$ on the infinite lattice $\mathbb{Z}$ at level $L$ is given as follows ($L = 2, 3, \ldots$ for the contact process and $L = 3, 4, \ldots$ for the asynchronous Domany-Kinzel model):

**Definition 1.** *Given the objective function $q(s) \in P_L$, $LP_{inv}(L)$ is a LP problem where*

- **Variables.** *Variables are $\left\langle \prod_{i \in A} s_i \right\rangle \in \mathbb{R}$, where $A \subset D_L$.*
- **Objective.** *Minimize the objective $\langle q(s) \rangle$ subject to the following constraints:*

*1. **Linearity.** Given any polynomials $q_1 \in P_L$ and $q_2 \in P_L$, with $\alpha \in \mathbb{R}$, their expectation values satisfy linearity: $\langle q_1 + \alpha q_2 \rangle = \langle q_1 \rangle + \alpha \langle q_2 \rangle$.*
*2. **Unit normalization.** $\langle 1 \rangle = 1$.*
*3. **Symmetry.** For any $A \subset D_L$ and $B \subset D_L$ such that $A \sim B$, $\left\langle \prod_{i \in A} s_i \right\rangle = \left\langle \prod_{i \in B} s_i \right\rangle$.*
*4. **Invariance.** For any polynomial $f(s) \in P_{L-1}$,*

$$\sum_{i \in D_{L-1}} \left\langle c(i, s) \left( f(\bar{s}^i) - f(s) \right) \right\rangle = 0 \,, \tag{32}$$

where $\left\langle \prod_{i \in A} s_i \right\rangle$ with $0 \in A$ is replaced by $\left\langle \prod_{i \in \tau_+(A)} s_i \right\rangle$ so that (32) closes within the variables under consideration.

**5. Probability bound.** *For any given spin assignment* $u \in \{1, -1\}^{D_L}$,

$$\left\langle \prod_{i \in D_L} \frac{1 + u_i s_i}{2} \right\rangle \geq 0. \tag{33}$$

*The minimum of* $\langle q(s) \rangle$ *obtained by* $LP_{inv}(L)$ *will be denoted as* $\langle q \rangle_L^{min}$.

Equations (32) are indeed the invariance equations (4) for $f(s) \in P_{L-1}$, where the infinite sum $\sum_{i \in \mathbb{Z}}$ truncates to a finite sum $\sum_{i \in D_{L-1}}$ since $f(\bar{s}^i) - f(s) = 0$ if $i \notin D_{L-1}$. The transition rates $c(i, s)$ for the contact process (5) and the asynchronous Domany-Kinzel model (7) involve spins $s_{i-1}, s_i$, and $s_{i+1}$, so that (32) may produce expectation values of functions depending on $s_0$, which are not in the space of variables $\left\langle \prod_{i \in A} s_i \right\rangle$ with $A \subset D_L$. We therefore make use of translation invariance to shift such functions so that (32) closes within the space of these variables.

The constraints of $LP_{inv}(L)$ form a proper subset of those of $LP_{inv}(L')$ for any $L' > L$. Therefore, $\langle q \rangle_L^{min}$ is a non-decreasing function of $L$. We can similarly define an LP hierarchy for maximizing the objective $\langle q(s) \rangle$ and obtain the corresponding maximum $\langle q \rangle_L^{max}$, which is a non-increasing function of $L$. These results provide rigorous bounds on the value of $\langle q(s) \rangle$ that *any* invariant measure, respecting the symmetries of the lattice, for the transition rate $c(i, s)$ on the infinite lattice $\mathbb{Z}$ must obey:

$$\langle q \rangle_L^{min} \leq \langle q(s) \rangle \leq \langle q \rangle_L^{max}, \quad \forall L. \tag{34}$$

Furthermore, **Theorem 5** in [9] implies that there exists an invariant measure respecting the symmetries whose expectation value of $q(s)$ agrees with the limiting value $\langle q \rangle_{L \to \infty}^{min}$, and similarly, there exists another invariant measure respecting the symmetries whose expectation value of $q(s)$ agrees with $\langle q \rangle_{L \to \infty}^{max}$. These two measures may or may not coincide.

## 2.2 Results for the contact process on $\mathbb{Z}$

We now present the bounds obtained from $LP_{inv}(L)$ for the contact process on $\mathbb{Z}$ with the transition rate (5). At low $L$, it is possible to obtain analytic expressions in terms of the rate $\lambda$, while at higher $L$, we use the LP solver `LinearOptimization` in *Mathematica* [47].

### 2.2.1 Analytic results at low $L$

We start with $L = 2$, assuming that $\lambda > 0$. After using the symmetry, the single invariance equation at this level is given by

$$1 - \lambda + \langle s_1 \rangle + \lambda \langle s_1 s_2 \rangle = 0 \quad \Rightarrow \quad \langle s_1 s_2 \rangle = -\frac{1 - \lambda + \langle s_1 \rangle}{\lambda}. \tag{35}$$

There are three linearly independent probability bounds, which are given by the following expressions after imposing the symmetries and (35):

$$-(1 - 2\lambda + \langle s_1 \rangle + 2\lambda \langle s_1 \rangle) \geq 0, \quad 1 + \langle s_1 \rangle \geq 0, \quad (2\lambda - 1)(1 + \langle s_1 \rangle) \geq 0. \tag{36}$$

Therefore, we find

$$\langle s_1 \rangle = -1, \quad \text{if } \lambda < \frac{1}{2}, \quad -1 \leq \langle s_1 \rangle \leq \frac{2\lambda - 1}{2\lambda + 1}, \quad \text{if } \lambda \geq \frac{1}{2}. \tag{37}$$

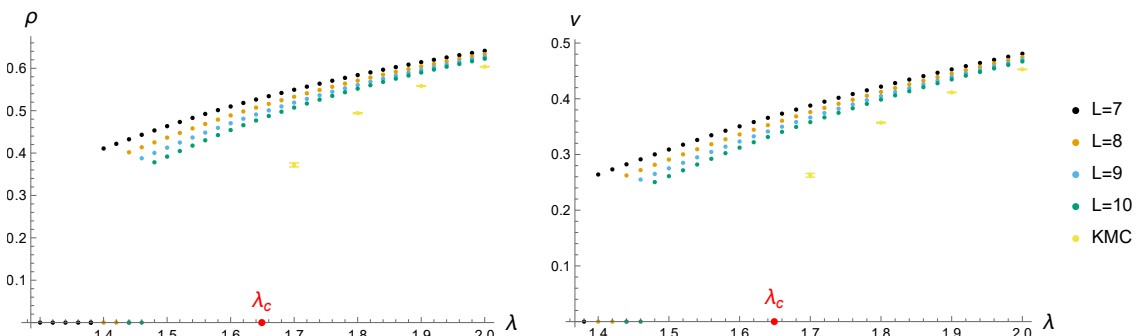

Figure 1: **Left**: $LP_{inv}(L)$ upper bounds on $\rho$ for the contact process on $\mathbb{Z}$ at $L = 7$ (black), $L = 8$ (orange), $L = 9$ (blue), and $L = 10$ (green), and also the KMC estimates (yellow) with $1\sigma$ error bars, which are hardly visible. The estimate for the critical rate $\lambda_c \approx 1.6491(1)$ from [40] is marked in red. **Right**: $LP_{inv}(L)$ upper bounds on $\nu$ for the contact process on $\mathbb{Z}$, together with the KMC estimates. Colors for data points are identical to those in the left figure.

Two conclusions can be made. The first equality implies a lower bound $\frac{1}{2} \leq \lambda_c$ on the critical rate $\lambda_c$ since $\langle s_1 \rangle = -1$ specifies the absorbing state. The second inequality shows an upper bound on $\langle s_1 \rangle$, or equivalently, an upper bound on the infection density $\rho \leq \frac{2\lambda}{2\lambda+1}$ of the nontrivial upper invariant measure for $\lambda > \lambda_c$. Note that the lower bound on $\rho$ is always given by $0 \leq \rho$ due to the presence of the absorbing state.

It is straightforward to extend the analysis to $L = 3$, leading to

$$\langle s_1 \rangle = -1, \quad \text{if } \lambda < 1, \qquad -1 \leq \langle s_1 \rangle \leq \frac{2\lambda^2 - \lambda - 1}{2\lambda^2 + \lambda + 1}, \quad \text{if } \lambda \geq 1, \tag{38}$$

implying $1 \leq \lambda_c$ and $0 \leq \rho \leq \frac{2\lambda^2}{2\lambda^2 + \lambda + 1}$ for $\lambda > \lambda_c$. The case $L = 4$ can still be solved analytically, providing $\frac{1+\sqrt{37}}{6} \leq \lambda_c$ and upper bounds on $\rho$, which are no longer as simple to express as in the cases of $L = 2, 3$.

### 2.2.2 Exact results at higher $L$

Given a rational value of $\lambda$, we now maximize $\rho = \left\langle \frac{1+s_1}{2} \right\rangle$ in $LP_{inv}(L)$ (i.e. $q(s) = \frac{1+s_1}{2}$) at higher values of $L$ to derive rigorous upper bounds on $\rho$, using the `LinearOptimization` function in *Mathematica* [47]. For example, $LP_{inv}(L = 8)$ at $\lambda = 2$ produces $\rho \leq \frac{5716599354130854044092609142591744}{9027340680181721890466616314606815} \approx 0.63325$, while it produces $\rho \leq 0$ at $\lambda = 1.42$, implying $1.42 \leq \lambda_c$. Note that due to the presence of the absorbing state, minimization of $\rho$ in $LP_{inv}(L)$ always produces a trivial lower bound $0 \leq \rho$. Therefore, we discuss only the upper bounds on $\rho$.

We performed the analysis up to $L = 10$ and obtained the left plot of Figure 1, where the KMC estimates obtained from 200 independent simulations over a periodic lattice of size 200 are also shown (see Appendix B for more details on the KMC simulations). We observe that the upper bounds are closer to the KMC estimates at larger values of $\lambda$. **Theorem 5** in [9] implies that as $L \to \infty$, the upper bounds will converge to the value realized by the nontrivial upper invariant measure in the supercritical phase.

We also performed a similar analysis to obtain upper bounds on

$$\nu = \left\langle \frac{1+s_i}{2} \frac{1+s_{i+1}}{2} \right\rangle, \tag{39}$$

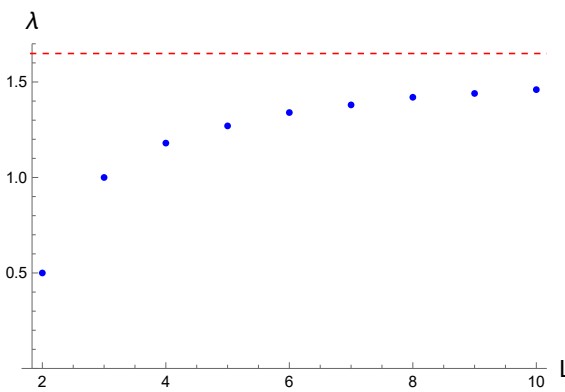

Figure 2: $LP_{inv}(L)$ lower bounds on $\lambda_c$ for the contact process on $\mathbb{Z}$ at different values of $L$ (blue dots). For comparison, the estimate $\lambda_c \approx 1.6491(1)$ is also shown (dotted red line).

which is the probability that two adjacent sites take spin values $+1$. $LP_{inv}(L)$ upper bounds on $\nu$ are presented in the right plot of Figure 1, together with the KMC estimates.

$LP_{inv}(L = 10)$ also provided the lower bound $1.46 \leq \lambda_c$ by obtaining $\rho \leq 0$ at $\lambda = 1.46$. Such lower bounds at different values of $L$ are presented in Figure 2. These results are weaker than the bound $1.5388 \leq \lambda_c$ obtained in [6], where auxiliary stochastic processes whose critical rates lower bound $\lambda_c$ are constructed based on monotonicity and coupling arguments.

### 2.3 Results for the asynchronous Domany-Kinzel model on $\mathbb{Z}$

To illustrate that bootstrap methods are applicable regardless of specific properties like monotonicity, we now apply $LP_{inv}(L)$ to the asynchronous Domany-Kinzel model with $p_2 = 0$. We expect that there exists $p_{1c}$ such that at $p_1 = p_{1c}$, the model undergoes a continuous phase transition corresponding to the DP universality class in 1+1 dimensions. KMC results in Appendix B suggest $p_{1c} \approx 0.908$.

#### 2.3.1 Analytic results at $L = 3$

$LP_{inv}(L = 3)$ has four variables after symmetries are imposed: $\langle s_1 \rangle$, $\langle s_1 s_2 \rangle$, $\langle s_1 s_3 \rangle$, and $\langle s_1 s_2 s_3 \rangle$. Two invariance equations are given by

$$\langle s_1 \rangle + \langle s_1 s_2 \rangle = 0, \qquad 1 - p_1 + \langle s_1 \rangle + p_1 \langle s_1 s_3 \rangle = 0. \tag{40}$$

Using these, the probability bounds lead to

$$\langle s_1 \rangle = -1, \quad \text{if } p_1 < \frac{1}{2}, \qquad -1 \leq \langle s_1 \rangle \leq \frac{2p_1 - 1}{2p_1 + 1}, \quad \text{if } \frac{1}{2} \leq p_1 \leq 1, \tag{41}$$

implying $\frac{1}{2} \leq p_{1c}$ and $\rho \leq \frac{2p_1}{2p_1 + 1}$ for $p_1 > p_{1c}$.

#### 2.3.2 Exact results at higher $L$

Given a rational value of $p_1$, we use the `LinearOptimization` function in *Mathematica* to maximize $\rho$ in $LP_{inv}(L)$. The results are presented in the left plot of Figure 3, together with the KMC estimates. $LP_{inv}(L = 10)$ provides the lower bound $0.839 \leq p_{1c}$ by obtaining $\rho \leq 0$ at $p_1 = 0.839$. Such lower bounds at different values of $L$ are shown in the right plot of Figure 3, consistent with the KMC estimate $p_{1c} \approx 0.908$.

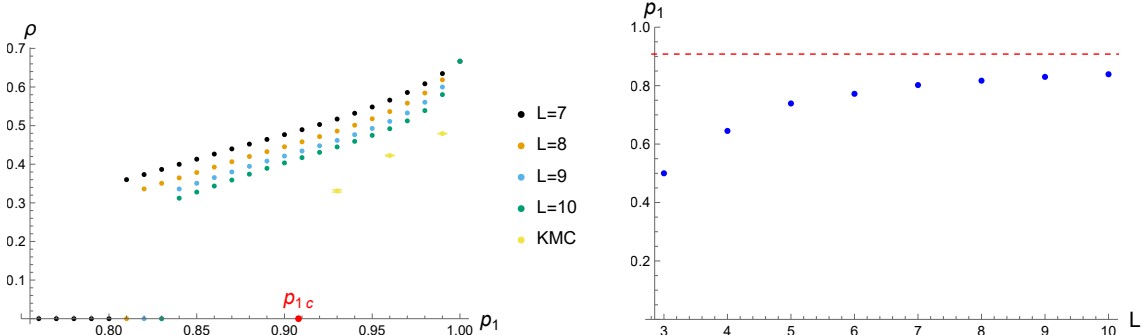

Figure 3: **Left**: $LP_{inv}(L)$ upper bounds on $\rho$ for the asynchronous Domany-Kinzel model at $L = 7$ (black), $L = 8$ (orange), $L = 9$ (blue), and $L = 10$ (green), and also the KMC estimates (yellow) with $1\sigma$ error bars, which are hardly visible. The estimate for the critical rate $p_{1c} \approx 0.908$ is marked in red. **Right**: $LP_{inv}(L)$ lower bounds on $p_{1c}$ for the asynchronous Domany-Kinzel model at different values of $L$ (blue dots). For comparison, the estimate $p_{1c} \approx 0.908$ is also presented (dotted red line).

## 2.4  LP hierarchy for the contact process on $\mathbb{Z}^2$

Bootstrap methods can be straightforwardly extended to higher dimensions. We consider the contact process on $\mathbb{Z}^2$ in this section. As already mentioned, there is a critical rate $\lambda_{c,2}$ such that for $\lambda \leq \lambda_{c,2}$, the absorbing state is the unique invariant measure, while for $\lambda > \lambda_{c,2}$, there exists a nontrivial upper invariant measure. The contact process on $\mathbb{Z}^2$ belongs to the DP universality class in 2+1 dimensions.

The lattice symmetry group $\mathbb{Z}^2 \rtimes D_4$ respected by the process is generated by lattice translations along the $x$- and $y$-directions, $\frac{\pi}{2}$-rotation around the origin, and reflection about the $x$-axis. Similarly to the case of $\mathbb{Z}$, we denote $A \sim B$ for two finite subsets $A$ and $B$ of $\mathbb{Z}^2$ if they can be mapped to each other via these symmetry actions. We focus on the measures that respect the full symmetries by imposing the corresponding equalities for the expectation values, similar to constraint 3 of $LP_{inv}(L)$.

To systematize the LP hierarchy for the contact process on $\mathbb{Z}^2$, we define $\tilde{D}_L = \{i \in \mathbb{Z}^2 \mid \|i\|_1 \leq L-1\}$, where $\|\cdot\|_1$ is the $L_1$-norm and $L = 1, 2, \ldots$. For each $j \in \partial\tilde{D}_{L+1} = \{i \in \mathbb{Z}^2 \mid \|i\|_1 = L\}$, we define $\bar{D}_L^j = \tilde{D}_L \cup \{j\}$. The idea is that if we take $f(s)$ in the invariance equation (4) to depend only on the spins over $\tilde{D}_L$, then the equation depends on the expectation values of functions that depend only on the spins over $\bar{D}_L^j$ for $j \in \partial\tilde{D}_{L+1}$.

The variables of the LP at level $L$, denoted as $LP_{inv}^{2d}(L)$, are now expectation values of spins over $\bar{D}_L^j$ for $j \in \partial\tilde{D}_{L+1}$, with the constraints being the invariance equations and probability bounds that close within them. Finally, define $\tilde{P}_L$ and $P_L'$ to be the sets of polynomials of spins over $\tilde{D}_L$ and $\bar{D}_L^j$ for $j \in \partial\tilde{D}_{L+1}$, respectively.

**Definition 2.** *Given the objective function $q(s) \in P_L'$, $LP_{inv}^{2d}(L)$ is a LP problem where*

- **Variables.** *Variables are $\left\langle \prod_{i \in A} s_i \right\rangle \in \mathbb{R}$, where $A \subset \bar{D}_L^j$ for $j \in \partial\tilde{D}_{L+1}$.*
- **Objective.** *Minimize (or maximize) the objective $\langle q(s) \rangle$ subject to the following constraints:*

**1. Linearity.** *Given any polynomials $q_1 \in P_L'$ and $q_2 \in P_L'$, with $\alpha \in \mathbb{R}$, their expectation values satisfy linearity: $\langle q_1 + \alpha q_2 \rangle = \langle q_1 \rangle + \alpha \langle q_2 \rangle$.*
**2. Unit normalization.** $\langle 1 \rangle = 1$.
**3. Symmetry.** *For any $A \subset \bar{D}_L^j$ and $B \subset \bar{D}_L^k$ for $j, k \in \partial\tilde{D}_{L+1}$ such that $A \sim B$,*

$$\left\langle \prod_{i \in A} s_i \right\rangle = \left\langle \prod_{i \in B} s_i \right\rangle.$$

**4. Invariance.** *For any polynomial* $f(s) \in \tilde{P}_L$,

$$\sum_{i \in \tilde{D}_L} \left\langle c(i,s)\left(f(\bar{s}^i) - f(s)\right) \right\rangle = 0, \tag{42}$$

*where* $c(i,s)$ *is given by* (5) *with* $d = 2$.

**5. Probability bound.** *For each* $j \in \partial \tilde{D}_{L+1}$, *and any given spin assignment* $u \in \{1,-1\}^{\tilde{D}_L^j}$,

$$\left\langle \prod_{i \in \tilde{D}_L^j} \frac{1 + u_i s_i}{2} \right\rangle \geq 0. \tag{43}$$

As before, the obtained minimum (maximum) provides a rigorous lower (upper) bound on the value of $\langle q(s) \rangle$ realized by any invariant measure respecting the lattice symmetries. $LP_{inv}^{2d}(L=1)$ has only two variables, $\langle s_{(0,0)} \rangle$ and $\langle s_{(0,0)} s_{(1,0)} \rangle$, after imposing all the symmetries. The invariance equation leads to

$$\langle s_{(0,0)} s_{(1,0)} \rangle = \frac{2\lambda - 1 - \langle s_{(0,0)} \rangle}{2\lambda}, \tag{44}$$

and the probability bounds, after imposing (44), are given by

$$1 + \langle s_{(0,0)} \rangle \geq 0, \qquad 4\lambda - 1 - (4\lambda + 1)\langle s_{(0,0)} \rangle \geq 0, \qquad (4\lambda - 1)(1 + \langle s_{(0,0)} \rangle) \geq 0. \tag{45}$$

Therefore, we conclude

$$\langle s_{(0,0)} \rangle = -1, \quad \text{if } \lambda < \frac{1}{4}, \qquad -1 \leq \langle s_{(0,0)} \rangle \leq \frac{4\lambda - 1}{4\lambda + 1}, \quad \text{if } \lambda \geq \frac{1}{4}, \tag{46}$$

implying a lower bound $\frac{1}{4} \leq \lambda_{c,2}$ on the critical rate $\lambda_{c,2}$ for the contact process on $\mathbb{Z}^2$. The numerical estimate from Monte Carlo simulations is given by $\lambda_{c,2} \approx 0.41220(3)$ [49].

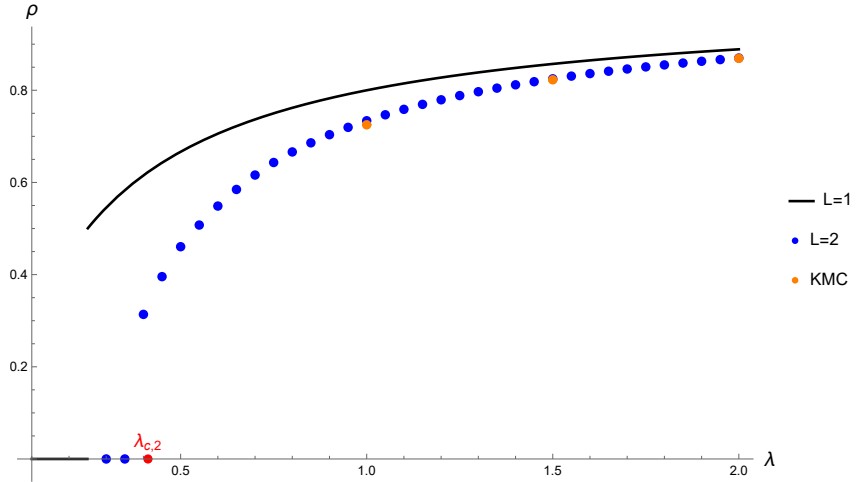

Figure 4: $LP_{inv}^{2d}(L)$ upper bounds on $\rho$ for the contact process on $\mathbb{Z}^2$ at $L = 1$ (black line) and $L = 2$ (blue dots), and KMC estimates (orange dots) with $1\sigma$ error bars, which are hardly visible. The Monte Carlo estimate for $\lambda_{c,2} \approx 0.41220(3)$ from [49] is marked in red.

Table 1: Comparisons between $LP_{inv}^{2d}(L=2)$ upper bounds and KMC estimates for $\rho$.

| $\lambda$ | $LP_{inv}^{2d}(L=2)$ | KMC |
|---|---|---|
| 1 | $\frac{1915290}{2610007} \approx 0.733826$ | 0.72506(26) |
| 1.5 | $\frac{306738522}{371876605} \approx 0.824840$ | 0.82278(17) |
| 2 | $\frac{3540162784}{4069310825} \approx 0.869966$ | 0.86931(11) |

We also obtained upper bounds on the infection density $\rho = \left\langle \frac{1+s_{(0,0)}}{2} \right\rangle$ from $LP_{inv}^{2d}(L=2)$ using the `LinearOptimization` function in *Mathematica*. The results are presented in Figure 4. In particular, at $\lambda = 0.362$, we obtain $\rho \leq 0$, implying a lower bound $0.362 \leq \lambda_{c,2}$ on the critical rate. In Table 1, explicit numerical comparisons between exact upper bounds on $\rho$ obtained from $LP_{inv}^{2d}(L=2)$ and KMC estimates for $\rho$ are presented. We observe that as $\lambda$ increases, the difference between the two decreases.

## 2.5 LP hierarchy for the synchronous Domany-Kinzel model on $\mathbb{Z}$

The LP hierarchy for the synchronous Domany-Kinzel model on $\mathbb{Z}$ is very much analogous to $LP_{inv}(L)$, with the same set of symmetry constraints. Using the same notations as before, $LP_{inv}^s(L)$ for $L = 2, 3, \ldots$ is a hierarchy of LPs for the invariant measures of the synchronous Domany-Kinzel model, defined as follows:

**Definition 3.** *Given the objective function $q(s) \in P_L$, $LP_{inv}^s(L)$ is a LP problem where*

- **Variables.** *Variables are $\left\langle \prod_{i \in A} s_i \right\rangle \in \mathbb{R}$, where $A \subset D_L$.*
- **Objective.** *Minimize the objective $\langle q(s) \rangle$ subject to the following constraints:*

**1. Linearity.** *Given any polynomials $q_1 \in P_L$ and $q_2 \in P_L$, with $\alpha \in \mathbb{R}$, their expectation values satisfy linearity: $\langle q_1 + \alpha q_2 \rangle = \langle q_1 \rangle + \alpha \langle q_2 \rangle$.*
**2. Unit normalization.** $\langle 1 \rangle = 1$.
**3. Symmetry.** *For any $A \subset D_L$ and $B \subset D_L$ such that $A \sim B$, $\left\langle \prod_{i \in A} s_i \right\rangle = \left\langle \prod_{i \in B} s_i \right\rangle$.*
**4. Invariance.** *For any polynomial $f(s) \in P_{L-1}$,*

$$\langle f(s) \rangle = \sum_{A \subset D_L} C_A(f) \left\langle \prod_{i \in A} s_i \right\rangle. \tag{47}$$

**5. Probability bound.** *For any given spin assignment $u \in \{1, -1\}^{D_L}$,*

$$\left\langle \prod_{i \in D_L} \frac{1 + u_i s_i}{2} \right\rangle \geq 0. \tag{48}$$

Recall that $C_A(f)$ are real coefficients completely determined by the update rule (8). The update rule provides the list of probabilities for all spin configurations over $D_L$, which can then be expressed as the expectation values of the corresponding indicator functions, as explained below (12).

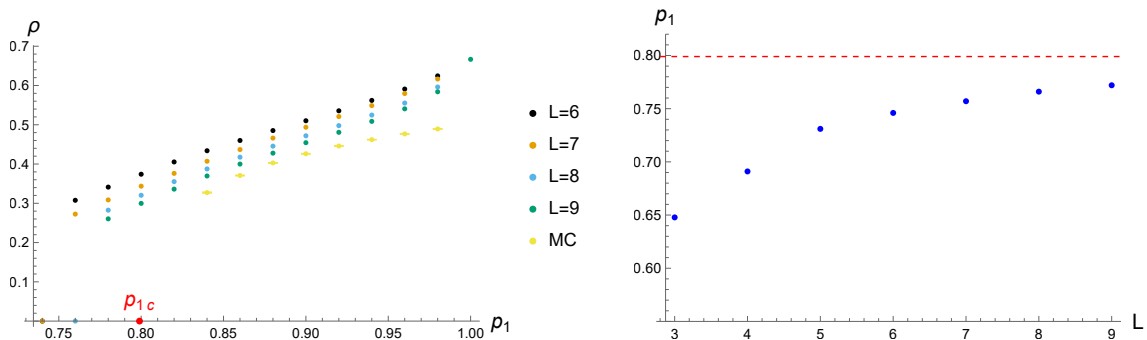

Figure 5: **Left**: $LP^s_{inv}(L)$ upper bounds on $\rho$ for the synchronous Domany-Kinzel model at $L = 6$ (black), $L = 7$ (orange), $L = 8$ (blue), and $L = 9$ (green), and also the Monte Carlo estimates (yellow) obtained by averaging over 200 independent simulations on a periodic lattice of size 201 for 400 time steps. Monte Carlo $1\sigma$ error bars are hardly visible. The estimate for the critical rate $p_{1c} \approx 0.799$ [50] is marked in red. **Right**: $LP^s_{inv}(L)$ lower bounds on $p_{1c}$ at different values of $L$ (blue dots). For comparison, the estimate $p_{1c} \approx 0.799$ [50] is also presented (dotted red line).

We now consider $LP^s_{inv}(L)$ for the case of Wolfram's rule 18, where $p_2 = 0$, which is a non-monotonic process. At $L = 3$, for example, after symmetries are imposed, the variables are the expectation values of $s_1, s_1 s_2, s_1 s_3$, and $s_1 s_2 s_3$. The invariance equations then lead to the following relations among them:

$$\langle s_1 s_2 \rangle = 1 - \frac{1}{p_1} - \frac{\langle s_1 \rangle}{p_1}, \qquad \langle s_1 s_3 \rangle = \frac{2p_1 - 2p_1^2 + p_1^3 - 1 + (2p_1 - 2p_1^2 - 1)\langle s_1 \rangle}{p_1^3}. \tag{49}$$

Combined with probability bounds, they lead to the lower bound $x_* \le p_{1c}$ on the critical value $p_{1c}$ for $p_1$, where $x_* \approx 0.64780$ is the unique real solution to the equation $2x^3 - 2x^2 + 2x - 1 = 0$.

Results for higher $L$ are presented in the left plot of Figure 5. At $L = 9$, we obtain $0.772 \le p_{1c}$, consistent with the estimate $0.799(2)$ from [50]. Lower bounds on $p_{1c}$ at different values of $L$ are presented in the right plot of Figure 5.

## 2.6 LP hierarchy for the upper invariant measure

The bootstrap bounds discussed in the previous subsections apply to arbitrary invariant measures. However, it is desirable to derive bounds that apply specifically to the upper invariant measure in the supercritical phase. In this subsection, we discuss such bounds on ratios among the expectation values of the upper invariant measure. For concreteness, we focus on the lattice $\mathbb{Z}$; the generalization to higher dimensions is straightforward.

As introduced in section 1.2.1, we consider the ratios

$$\mathcal{R}\left(\prod_{i \in A} s_i\right) = \frac{\mathcal{D}\left(\prod_{i \in A} s_i\right)}{\mathcal{D}(s_i)} = \frac{\left\langle \prod_{i \in A} s_i \right\rangle^{up} - (-1)^{|A|}}{\langle s_i \rangle^{up} + 1}. \tag{50}$$

Both the asynchronous and synchronous invariance equations (4) and (11) can be reformulated as linear equations in $\mathcal{R}$. Invariance equations that close within the expectation values of polynomials in $P_L$, under the assumption of translation invariance, can be expressed as

$$\sum_{A \subset D_L} \mathcal{U}^\kappa_{L,A} \mathcal{R}\left(\prod_{i \in A} s_i\right) = 0, \quad \forall \kappa \in \Upsilon_L, \tag{51}$$

where $\Upsilon_L$ is an appropriate finite discrete index set for each $L$, and $\mathcal{U}_{L,A}^{\kappa} \in \mathbb{R}$ is determined by the specific invariance equation under consideration. Combined with the probability bounds (19) on $\mathcal{R}$, we arrive at the following LP problem.

**Definition 4.** *Given the objective function $q(s) \in P_L$, $LP_{up}(L)$ is a LP problem where*
- **Variables.** *Variables are $\mathcal{R}\left(\prod_{i \in A} s_i\right) \in \mathbb{R}$, where $A \subset D_L$.*
- **Objective.** *Minimize the objective $\mathcal{R}(q(s))$ subject to the following constraints:*

**1. Linearity.** *Given any polynomials $q_1 \in P_L$ and $q_2 \in P_L$, with $\alpha \in \mathbb{R}$, their $\mathcal{R}$ values satisfy linearity: $\mathcal{R}(q_1 + \alpha q_2) = \mathcal{R}(q_1) + \alpha \mathcal{R}(q_2)$.*
**2. Difference and normalization.** *$\mathcal{R}(1) = 0$, $\mathcal{R}(s_i) = 1$.*
**3. Symmetry.** *For any $A \subset D_L$ and $B \subset D_L$ such that $A \sim B$, $\mathcal{R}\left(\prod_{i \in A} s_i\right) = \mathcal{R}\left(\prod_{i \in B} s_i\right)$.*
**4. Invariance.** *For each $\kappa \in \Upsilon_L$,*

$$\sum_{A \subset D_L} \mathcal{U}_{L,A}^{\kappa} \mathcal{R}\left(\prod_{i \in A} s_i\right) = 0. \tag{52}$$

**5. Probability bound.** *For any given spin assignment $u \in \{1, -1\}^{D_L}$ such that $\exists i \in D_L$ with $u_i = 1$,*

$$\mathcal{R}\left(\prod_{i \in D_L} \frac{1 + u_i s_i}{2}\right) \geq 0. \tag{53}$$

In formulating $LP_{up}(L)$, we have assumed the existence of the upper invariant measure at a given value of $\lambda$. If $\lambda < \lambda_c$, the result of $LP_{up}(L)$ is void. In contrast, when $\lambda > \lambda_c$, the result becomes nontrivial.

### 2.6.1 Results for the contact process on $\mathbb{Z}$

Consider the contact process on $\mathbb{Z}$. At $L = 2$, substituting (16) into (35) yields

$$\mathcal{D}(s_1) + \lambda \mathcal{D}(s_1 s_2) = 0, \tag{54}$$

which leads to

$$\frac{\langle s_1 s_2 \rangle^{up} - 1}{\langle s_1 \rangle^{up} + 1} = \frac{\mathcal{D}(s_1 s_2)}{\mathcal{D}(s_1)} = \mathcal{R}(s_1 s_2) = -\frac{1}{\lambda}, \tag{55}$$

thus completely determining the ratio.

At $L = 3$, there is one additional invariance equation, which simplifies to

$$\mathcal{R}(s_1 s_2 s_3) = 1 - \frac{2}{\lambda^2} + \mathcal{R}(s_1 s_3). \tag{56}$$

Combined with the probability bounds, we obtain

$$-\frac{1 + \lambda}{\lambda^2} \leq \mathcal{R}(s_1 s_3) \leq -\frac{1}{\lambda^2}, \quad \text{for } \lambda \geq 1. \tag{57}$$

Bounds on $\mathcal{R}(s_1 s_3)$ obtained from $LP_{up}(L)$ using `LinearOptimization` at $L = 5, 6$, and 7 are presented in Figure 6. At $\lambda = 2$, $LP_{up}(L = 10)$ yields the bounds given in (28).

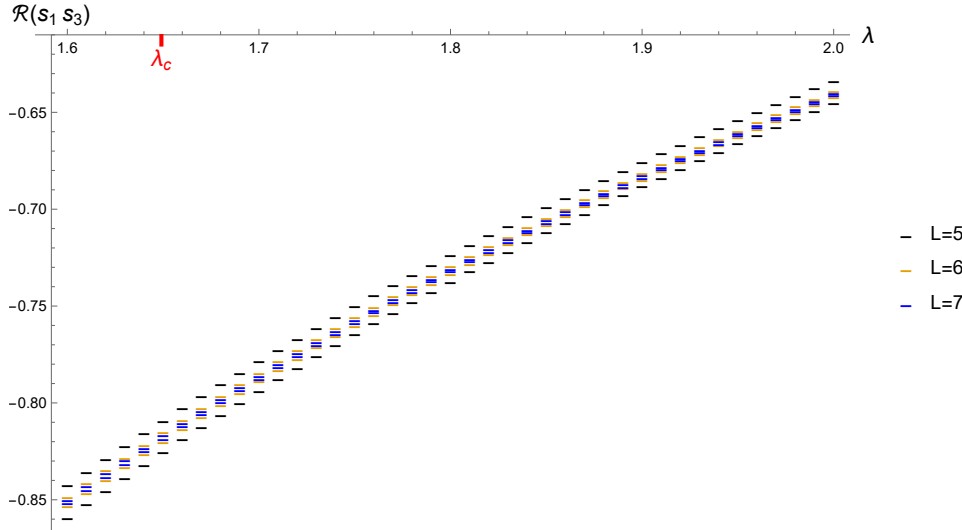

Figure 6: $LP_{up}(L)$ upper and lower bounds on $\mathcal{R}(s_1 s_3)$ for the contact process on $\mathbb{Z}$ at $L = 5$ (black), $L = 6$ (orange), and $L = 7$ (blue). The upper and lower bounds at $L = 7$ are nearly indistinguishable.

### 2.6.2 Results for the synchronous Domany-Kinzel model on $\mathbb{Z}$

The formulation $LP_{up}(L)$ also applies to synchronous processes. For the synchronous Domany-Kinzel model at $p_2 = 0$, the invariance equations at $L = 3$, for example, imply

$$\mathcal{R}(s_1 s_2) = -\frac{1}{p_1}, \qquad \mathcal{R}(s_1 s_3) = -\frac{1 - 2p_1 + 2p_1^2}{p_1^3}. \qquad (58)$$

On the other hand, $\mathcal{R}(s_1 s_2 s_3)$ is not completely determined. At $p_1 = 0.9$, $LP_{up}(L = 9)$ yields

$$1.0617 < \mathcal{R}(s_1 s_2 s_3) < 1.1989, \qquad (59)$$

which is consistent with the Monte Carlo estimate $\mathcal{R}(s_1 s_2 s_3) \approx 1.1364(12)$, obtained from 200 independent simulations with random initial configurations on a periodic lattice of size 201 evolved over 400 time steps.

## 3 Bootstrapping the short-time evolution of noninvariant measures

In this section, we introduce bootstrap methods for deriving bounds on the short-time evolution of the expectation values of noninvariant measures in asynchronous processes. We start by noting a crucial distinction between synchronous and asynchronous time evolutions, which explains why bootstrap methods are desirable for the asynchronous case but not necessary for the synchronous case.

Consider the synchronous Domany-Kinzel model on $\mathbb{Z}$, whose expectation values obey discrete time evolution equations (9). In order to determine $\langle s_i(t+1) \rangle$, for example, (10) implies that we need the values of $\langle s_i(t) \rangle$, $\langle s_{i+1}(t) \rangle$, and $\langle s_i(t)s_{i+1}(t) \rangle$. In general, to determine $\langle f(s(t+1)) \rangle$ for $f(s) \in P_L$, we need the values of $\langle g(s(t)) \rangle$ for $g(s) \in P_{L+1}$. Therefore, $\langle f(s(t=T)) \rangle$ for a given $f(s) \in P_L$ and $T \in \mathbb{N}$ can be determined if we are given the initial values $\langle g(s(t=1)) \rangle$ for all $g(s) \in P_{T+L-1}$. We explore this idea further in section 3.8.

In contrast, the master equation (3) for asynchronous processes is a differential equation involving time derivatives. For example, in $d = 1$, $\frac{d}{dt}\langle f(s)\rangle_t$ for $f(s) \in P_L$ is equal to a linear combination of the expectation values $\langle g(s)\rangle_t$ for $g(s) \in P_{L+1}$ (assuming translation invariance). Since such differential equations do not contain information about how $\langle \tilde{g}(s)\rangle_t$ for $\tilde{g}(s) \notin P_L$, $\tilde{g}(s) \in P_{L+1}$ evolves in time, its appearance in the master equation for $\frac{d}{dt}\langle f(s)\rangle_t$ makes the equation not explicitly solvable, even if all the initial conditions are specified. Instead, it is still possible to *bound* $\langle f(s)\rangle_{t=T}$ at any fixed finite $T \in \mathbb{R}$ using the bootstrap methods.

## 3.1 Primal optimization problem

Bounding time-dependent objectives governed by differential equations such as (3) using convex optimization is a well-established problem in optimal control theory (see, e.g., [12, 18]), and has recently been extended to the time evolution of expectation values in quantum systems [13, 14]. We now apply this approach to asynchronous stochastic processes.

For concreteness, we focus on $d = 1$. Our goal is to find lower and upper bounds on $\langle q(s)\rangle_{t=T}$ given some initial conditions on the expectation values at $t = 0$ that respect the lattice symmetries. It is straightforward to formulate the optimization problem based on the master equation (3). The corresponding level $L$ *primal* optimization problem $PO(L)$ is defined as follows:

**Definition 5.** *Given the objective function $q(s) \in P_{L-1}$, initial conditions $\left\langle \prod_{i \in A} s_i \right\rangle_{t=0} = y_A$ for $A \in D_L$ respecting the lattice symmetries, and time $T > 0$, $PO(L)$ is a primal optimization problem where*

- **Variables.** *Variables are class $C^1$ functions $\left\langle \prod_{i \in A} s_i \right\rangle_t$ of $t \in [0, T]$, where $A \subset D_L$.*
- **Objective.** *Minimize the objective $\langle q(s)\rangle_{t=T}$ subject to the following constraints:*

**1. Linearity.** *Given any polynomials $q_1 \in P_L$ and $q_2 \in P_L$, with $\alpha \in \mathbb{R}$, their expectation values satisfy linearity: $\langle q_1 + \alpha q_2\rangle_t = \langle q_1\rangle_t + \alpha\langle q_2\rangle_t$ for $t \in [0, T]$.*
**2. Unit normalization.** *$\langle 1\rangle_t = 1$ for $t \in [0, T]$.*
**3. Symmetry.** *For any $A \subset D_L$ and $B \subset D_L$ such that $A \sim B$, $\left\langle \prod_{i \in A} s_i \right\rangle_t = \left\langle \prod_{i \in B} s_i \right\rangle_t$ for $t \in [0, T]$.*
**4. Master equation.** *For any polynomial $f(s) \in P_{L-1}$,*

$$\frac{d}{dt}\langle f(s)\rangle_t = \sum_{i \in D_{L-1}} \left\langle c(i, s)\left(f(\bar{s}^i) - f(s)\right)\right\rangle_t, \quad t \in [0, T], \tag{60}$$

*where $\left\langle \prod_{i \in A} s_i \right\rangle_t$ with $0 \in A$ is replaced by $\left\langle \prod_{i \in \tau_+(A)} s_i \right\rangle_t$.*
**5. Probability bound.** *For any given spin assignment $u \in \{1, -1\}^{D_L}$,*

$$\left\langle \prod_{i \in D_L} \frac{1 + u_i s_i}{2} \right\rangle_t \geq 0, \quad t \in [0, T]. \tag{61}$$

**6. Initial condition.** *$\left\langle \prod_{i \in A} s_i \right\rangle_{t=0} = y_A$ for $A \in D_L$.*

*The minimum of $\langle q(s)\rangle_{t=T}$ obtained by $PO(L)$ will be denoted as $\langle q\rangle_{t=T}^{min,L}$.*

### 3.2 Dual optimization problem

In $PO(L)$, even though $L$ is finite, the space of variables is infinite-dimensional since they are functions of $t$. Therefore, it may not be immediately obvious how to find the minimum $\langle q \rangle_{t=T}^{min,L}$ over such a space. However, the standard weak duality theorem in optimization implies that any feasible solution of the *dual* optimization problem produces a lower bound on $\langle q \rangle_{t=T}^{min,L}$, which would then serve as a lower bound also on the actual value of $\langle q(s) \rangle_{t=T}$.

We introduce a modified version of $PO(L)$ before turning to the dual problem. First, we explicitly solve constraint 3 of $PO(L)$ and substitute the solutions into the variables, objective, and constraints 4, 5, and 6 of $PO(L)$, using constraints 1 and 2. Denote by $X_t^a$, $a = 1, \ldots, m$, the independent variables remaining after this procedure.

**Definition 6.** *The primal optimization problem $PO'(L)$ at level $L$ equivalent to $PO(L)$ is given by the following:*
- ***Variables.*** *Variables are class $C^1$ functions $X_t^a$ of $t \in [0, T]$, where $a \in \{1, \ldots, m\}$.*
- ***Objective.*** *Minimize the objective $Q + \sum_{a=1}^m Q^a X_{t=T}^a$ subject to the following constraints:*

**1. Master equation.** *For $\beta = 1, \ldots, n$,*

$$r_\beta + \sum_{a=1}^m \left( W_\beta^a \frac{d}{dt} + V_\beta^a \right) X_t^a = 0, \quad t \in [0, T]. \tag{62}$$

**2. Probability bound.** *For $\gamma = 1, \ldots, l$,*

$$h_\gamma + \sum_{a=1}^m G_\gamma^a X_t^a \geq 0, \quad t \in [0, T]. \tag{63}$$

**3. Initial condition.** *For $a = 1, \ldots, m$,*

$$X_{t=0}^a = g^a. \tag{64}$$

*The minimum obtained by $PO'(L)$ is the same as $\langle q \rangle_{t=T}^{min,L}$.*

Here, $m, n, l \in \mathbb{N}$ and $Q, Q^a, r_\beta, W_\beta^a, V_\beta^a, h_\gamma, G_\gamma^a, g^a \in \mathbb{R}$ are completely determined by the ingredients of $PO(L)$. We now formulate the dual problem $DO(L)$ of the primal problem $PO'(L)$. As usual, we introduce the Lagrange multipliers $\lambda_W^\beta(t)$, $\lambda_G^\gamma(t)$, and $\lambda_g^a$ for constraints 1, 2, and 3 of $PO'(L)$, respectively. The functions $\lambda_W^\beta(t)$ and $\lambda_G^\gamma(t)$ are class $C^1$ and $C^0$ functions of $t \in [0, T]$, respectively, while $\lambda_g^a \in \mathbb{R}$. After varying the Lagrangian with respect to the primal variables $X_t^a$, we arrive at the dual optimization problem.

**Definition 7.** *The dual optimization problem $DO(L)$ of the primal optimization problem $PO'(L)$ is given by the following:*
- ***Variables.*** *Variables are class $C^1$ functions $\lambda_W^\beta(t)$ and class $C^0$ functions $\lambda_G^\gamma(t)$ of $t \in [0, T]$, and $\lambda_g^a \in \mathbb{R}$.*
- ***Objective.*** *Maximize the objective*

$$Q - \sum_{a=1}^m g^a \lambda_g^a + \int_0^T dt \left( \sum_{\beta=1}^n r_\beta \lambda_W^\beta(t) - \sum_{\gamma=1}^l h_\gamma \lambda_G^\gamma(t) \right), \tag{65}$$

*subject to the following constraints:*

**1. Primal objective.** *For $a = 1, \ldots, m$,*

$$Q^a + \sum_{\beta=1}^n \lambda_W^\beta(t = T) W_\beta^a = 0. \tag{66}$$

**2. Dual master equation.** *For* $a = 1, \ldots, m$,

$$\sum_{\beta=1}^{n} \left( -\frac{d\lambda_W^\beta(t)}{dt} W_\beta^a + \lambda_W^\beta(t) V_\beta^a \right) - \sum_{\gamma=1}^{l} \lambda_G^\gamma(t) G_\gamma^a = 0, \quad t \in [0, T]. \tag{67}$$

**3. Dual probability bound.** *For* $\gamma = 1, \ldots, l$,

$$\lambda_G^\gamma(t) \geq 0, \quad t \in [0, T]. \tag{68}$$

**4. Dual initial condition.** *For* $a = 1, \ldots, m$,

$$\lambda_g^a - \sum_{\beta=1}^{n} \lambda_W^\beta(t=0) W_\beta^a = 0. \tag{69}$$

*The maximum value of the objective obtained by* $DO(L)$ *is denoted as* $\langle q \rangle_{t=T}^{min,DO(L)}$.

Weak duality theorem states that

$$\langle q \rangle_{t=T}^{min,DO(L)} \leq \langle q \rangle_{t=T}^{min,L}. \tag{70}$$

Furthermore, since $DO(L)$ is a maximization problem, *any* feasible solution to constraints 1, 2, 3, and 4 of $DO(L)$ provides a lower bound on $\langle q \rangle_{t=T}^{min,DO(L)}$. In other words, if some given $\lambda_W^\beta(t)$, $\lambda_G^\gamma(t)$, and $\lambda_g^a$ satisfy the constraints 1–4 of $DO(L)$, then

$$Q - \sum_{a=1}^{m} g^a \lambda_g^a + \int_0^T dt \left( \sum_{\beta=1}^{n} r_\beta \lambda_W^\beta(t) - \sum_{\gamma=1}^{l} h_\gamma \lambda_G^\gamma(t) \right) \leq \langle q \rangle_{t=T}^{min,DO(L)} \leq \langle q \rangle_{t=T}^{min,L}$$

$$\leq \langle q(s) \rangle_{t=T}, \tag{71}$$

thus obtaining a lower bound on the actual value of $\langle q(s) \rangle_{t=T}$. Upper bounds on $\langle q(s) \rangle_{t=T}$ can be similarly obtained by considering $PO(L)$ with objective $-\langle q(s) \rangle_{t=T}$ and modifying $DO(L)$ accordingly.

Since any feasible solution of $DO(L)$ provides a desired lower bound, we can consider a finite-dimensional subspace of the dual variables $\lambda_W^\beta(t)$ and $\lambda_G^\gamma(t)$ and search for feasible solutions within this subspace. Moreover, within the space of feasible solutions in the subspace, we can maximize the dual objective to obtain the best lower bound on $\langle q(s) \rangle_{t=T}$ attainable in the subspace. This results in a finite-dimensional optimization problem, which can be addressed using standard optimization solvers.

There is no canonical choice of subspace, but a desirable one is such that constraint 3 of $DO(L)$, which imposes positivity, can be naturally implemented. Therefore, subspaces where a natural positive function basis exists are advantageous, since one can expand constraint 3 in such a basis and impose nonnegativity of the expansion coefficients. In this work, we employ two types of such subspaces: B-splines and polynomials.

### 3.3 Dual optimization problem in B-spline basis

By definition, the clamped B-spline basis provides a positive function basis over the domain $t \in [0, T]$. We consider a uniform knot vector $v_{knot} = (t_0, t_1, \ldots, t_{N+D})$ with $t_0 = \ldots = t_D = 0$, $t_k = \frac{k-D}{N-D} T$ for $k = D+1, \ldots, N-1$, and $t_N = \ldots = t_{N+D} = T$ for $N$ independent degree-$D$ polynomial clamped splines $\phi_{k_D}^{(D)}(t)$ with $k_D = 1, \ldots, N$. Each $\phi_{k_D}^{(D)}(t)$ is nonnegative over $[0, T]$ and has support only on $[t_{k_D-1}, t_{k_D+D})$ within $[0, T]$.

Since the derivative of a clamped B-spline of degree $D$ is a linear combination of clamped B-splines of degree $D-1$ over the same knot vector $v_{knot}$, we consider the clamped B-spline bases of all degrees $\mu = 1, 2, \ldots, D$ over $v_{knot}$ so that constraint 2 of $DO(L)$ can be implemented.[13]

Discarding splines that have no support over the interior region $(t_D, t_N)$, we obtain a set of clamped B-spline basis functions $\mathcal{B}_D = \left\{ \phi_{k_{(\mu)}}^{(\mu)}(t) \,\middle|\, \mu = 1, \ldots, D, \ k_{(\mu)} = 1, \ldots, N+\mu-D \right\}$, where $\phi_{k_{(\mu)}}^{(\mu)}(t)$ is the $k_{(\mu)}$-th element of the degree-$\mu$ clamped B-spline basis over $v_{knot}$, nonnegative on $[0, T]$ and supported only on $[t_{k_{(\mu)}-1+D-\mu}, t_{k_{(\mu)}+D})$ within $[0, T]$. They satisfy

$$\phi_{k_{(\mu)}}^{(\mu)}(t) \geq 0, \quad \text{for } t \in [0, T], \qquad \phi_{k_{(\mu)}}^{(\mu)}(0) = \delta_{k_{(\mu)}, 1}, \qquad \phi_{k_{(\mu)}}^{(\mu)}(T) = \delta_{k_{(\mu)}, N+\mu-D}, \tag{72}$$

providing a desired positive function basis. Although $\mathcal{B}_D$ is not orthonormal, it is linearly independent. These basis elements can be conveniently generated using the BSplineBasis function in *Mathematica*.

Relations among the derivatives are given by

$$\frac{d}{dt} \phi_{k_{(\mu)}}^{(\mu)}(t) = \sum_{l_{(\mu-1)}=1}^{N+\mu-1-D} F_{k_{(\mu)} l_{(\mu-1)}}^{(\mu)} \phi_{l_{(\mu-1)}}^{(\mu-1)}(t), \quad \mu = 2, 3, \ldots, D, \tag{73}$$

where the expansion coefficients can be computed as

$$
\begin{aligned}
J_{k_{(\mu)}, l_{(\mu)}}^{(\mu)} &= \int_0^T dt\, \phi_{k_{(\mu)}}^{(\mu)}(t) \phi_{l_{(\mu)}}^{(\mu)}(t), & \mu &= 1, \ldots, D-1, \\
H_{k_{(\mu)}, l_{(\mu-1)}}^{(\mu)} &= \int_0^T dt \left( \frac{d\phi_{k_{(\mu)}}^{(\mu)}(t)}{dt} \right) \phi_{l_{(\mu-1)}}^{(\mu-1)}(t), & \mu &= 2, \ldots, D, \\
\Rightarrow F^{(\mu)} &= H^{(\mu)} \left( J^{(\mu-1)} \right)^{-1}, & \mu &= 2, \ldots, D.
\end{aligned}
\tag{74}
$$

We also define the integrals

$$w_{k_{(\mu)}}^{(\mu)} = \int_0^T dt\, \phi_{k_{(\mu)}}^{(\mu)}(t). \tag{75}$$

We expand $\lambda_W^\beta(t)$ and $\lambda_G^\gamma(t)$ in $\mathcal{B}_D$:

$$\lambda_W^\beta(t) = \sum_{\mu=2}^D \sum_{k_{(\mu)}=1}^{N+\mu-D} q_{(\mu, k_{(\mu)})}^\beta \phi_{k_{(\mu)}}^{(\mu)}(t), \qquad \lambda_G^\gamma(t) = \sum_{\mu=1}^D \sum_{k_{(\mu)}=1}^{N+\mu-D} p_{(\mu, k_{(\mu)})}^\gamma \phi_{k_{(\mu)}}^{(\mu)}(t). \tag{76}$$

Note that $\lambda_W^\beta$ does not include $\mu = 1$ basis elements since constraint 2 of $DO(L)$ involves $\frac{d\lambda_W^\beta}{dt}$. These expansions can be substituted into $DO(L)$, and each constraint can be written as a constraint on the expansion coefficients $q_{(\mu, k_{(\mu)})}^\beta$ and $p_{(\mu, k_{(\mu)})}^\gamma$. In particular, constraint 3 for the dual probability bound can be satisfied by imposing $p_{(\mu, k_{(\mu)})}^\gamma \geq 0$. This yields a finite-dimensional linear programming (LP) problem.

**Definition 8.** *The LP problem $DO_{sp}(L; D, N)$ obtained by using the clamped B-spline basis up to degree $D$ over the knot vector $v_{knot}$ defined above is given by the following:*
- **Variables.** *Variables are $\lambda_g^a$, $q_{(\mu, k_{(\mu)})}^\beta$, $p_{(\mu, k_{(\mu)})}^\gamma \in \mathbb{R}$ for $a = 1, \ldots, m$, $\beta = 1, \ldots, n$, $\gamma = 1, \ldots, l$,*

---

[13]We thank Barak Gabai, Henry Lin, and Zechuan Zheng for pointing out the need for the lower degree splines.

$\mu = 1, \ldots, D$, and $k_{(\mu)} = 1, \ldots, N + \mu - D$.

- **Objective.** *Maximize the objective*

$$Q - \sum_{a=1}^{m} g^a \lambda_g^a + \sum_{\beta=1}^{n} r_\beta \sum_{\mu=2}^{D} \sum_{k_{(\mu)}=1}^{N+\mu-D} q_{(\mu,k_{(\mu)})}^\beta w_{k_{(\mu)}}^{(\mu)} - \sum_{\gamma=1}^{l} h_\gamma \sum_{\mu=1}^{D} \sum_{k_{(\mu)}=1}^{N+\mu-D} p_{(\mu,k_{(\mu)})}^\gamma w_{k_{(\mu)}}^{(\mu)}, \tag{77}$$

*subject to the following constraints:*

**1. Primal objective.** *For $a = 1, \ldots, m$,*

$$Q^a + \sum_{\beta=1}^{n} W_\beta^a \sum_{\mu=2}^{D} q_{(\mu,N+\mu-D)}^\beta = 0. \tag{78}$$

**2. Dual master equation at degree D.** *For $a = 1, \ldots, m$ and $k_{(D)} = 1, \ldots, N$,*

$$\sum_{\beta=1}^{n} V_\beta^a q_{(D,k_{(D)})}^\beta - \sum_{\gamma=1}^{l} G_\gamma^a p_{(D,k_{(D)})}^\gamma = 0. \tag{79}$$

**3. Dual master equation at intermediate degrees.** *For $a = 1, \ldots, m$, $\mu = 2, \ldots, D-1$, and $k_{(\mu)} = 1, \ldots, N + \mu - D$,*

$$\sum_{\beta=1}^{n} \left( -W_\beta^a \sum_{l_{(\mu+1)}=1}^{N+\mu+1-D} q_{(\mu+1,l_{(\mu+1)})}^\beta F_{l_{(\mu+1)}k_{(\mu)}}^{(\mu+1)} + V_\beta^a q_{(\mu,k_{(\mu)})}^\beta \right) - \sum_{\gamma=1}^{l} G_\gamma^a p_{(\mu,k_{(\mu)})}^\gamma = 0. \tag{80}$$

**4. Dual master equation at degree 1.** *For $a = 1, \ldots, m$ and $k_{(1)} = 1, \ldots, N + 1 - D$,*

$$\sum_{\beta=1}^{n} W_\beta^a \sum_{l_{(2)}=1}^{N+2-D} q_{(2,l_{(2)})}^\beta F_{l_{(2)}k_{(1)}}^{(2)} + \sum_{\gamma=1}^{l} G_\gamma^a p_{(1,k_{(1)})}^\gamma = 0. \tag{81}$$

**5. Dual probability bound.** *For $\gamma = 1, \ldots, l$, $\mu = 1, \ldots, D$, and $k_{(\mu)} = 1, \ldots, N + \mu - D$,*

$$p_{(\mu,k_{(\mu)})}^\gamma \geq 0. \tag{82}$$

**6. Dual initial condition.** *For $a = 1, \ldots, m$,*

$$\lambda_g^a - \sum_{\beta=1}^{n} W_\beta^a \sum_{\mu=2}^{D} q_{(\mu,1)}^\beta = 0. \tag{83}$$

*The maximum value of the objective obtained by $DO_{sp}(L; D, N)$ is denoted as $\langle q \rangle_{t=T}^{min, DO_{sp}(L;D,N)}$.*

Any feasible solution to the constraints of $DO_{sp}(L; D, N)$ provides a feasible solution to those of $DO(L)$ via (76). Therefore,

$$\langle q \rangle_{t=T}^{min, DO_{sp}(L;D,N)} \leq \langle q \rangle_{t=T}^{min, DO(L)} \leq \langle q(s) \rangle_{t=T}, \tag{84}$$

as desired. As an illustration of $DO_{sp}(L; D, N)$, we apply it to the contact process on $\mathbb{Z}$ at $\lambda = 2$ with initial conditions $\left\langle \prod_{i \in A} s_i \right\rangle_{t=0} = 1$ for all $A \subset \mathbb{Z}$. Setting $T = \frac{1}{2}$, $DP_{sp}(3; 5, 10)$ implemented by the exact LP solver `LinearOptimization` in *Mathematica* produces

$$0.76045 \approx \frac{182376205152997}{239827079364600} \leq \rho_{t=\frac{1}{2}} \leq \frac{1337581726637467}{1671099742133240} \approx 0.80042. \tag{85}$$

### 3.4 Dual optimization problem in polynomial basis

Polynomials also provide another natural basis for positivity constraints, as demonstrated in numerous polynomial optimization problems. The Markov-Lukács theorem states that a univariate polynomial $z(t)$ is nonnegative over $t \in [0, T]$ if and only if it can be represented as

$$z(t) = z_1(t) + t(T - t)z_2(t), \tag{86}$$

where $z_1(t)$ and $z_2(t)$ are sums of squares. If $z(t)$ is of degree $2D$, then $z_1(t)$ and $z_2(t)$ are of maximal degrees $2D$ and $2(D-1)$, respectively. Introducing monomial basis vectors $v_1(t) = (1, t, t^2, \ldots, t^D)$ and $v_2(t) = (1, t, t^2, \ldots, t^{D-1})$, we can write $z_1(t) = v_1(t)^T Y_1 v_1(t)$ and $z_2(t) = v_2(t)^T Y_2 v_2(t)$, where $Y_1$ and $Y_2$ are real symmetric matrices of sizes $(D+1) \times (D+1)$ and $D \times D$, respectively. The statement that $z_1(t)$ and $z_2(t)$ are sums of squares is equivalent to the following positive semidefinite conditions:

$$Y_1 \succeq 0, \qquad Y_2 \succeq 0. \tag{87}$$

We therefore write

$$\lambda_G^\gamma(t) = v_1(t)^T Y_1^\gamma v_1(t) + t(T - t) v_2(t)^T Y_2^\gamma v_2(t), \qquad \lambda_W^\beta(t) = \sum_{k=0}^{2D} y_k^\beta t^k, \tag{88}$$

where $Y_1^\gamma$ and $Y_2^\gamma$ are real symmetric matrices of sizes $(D+1) \times (D+1)$ and $D \times D$, respectively, and $y_k^\beta \in \mathbb{R}$. Then, constraint 3 of $DO(L)$ is satisfied if

$$Y_1^\gamma \succeq 0, \qquad Y_2^\gamma \succeq 0. \tag{89}$$

By plugging (88) into $DO(L)$, we obtain a SDP problem, denoted by $DO_{poly}(L; D)$:

**Definition 9.** *The SDP problem $DO_{poly}(L; D)$ obtained by using the polynomial basis up to degree $2D$ is given by the following:*
- **Variables.** *Variables are real symmetric matrices $Y_1^\gamma$ and $Y_2^\gamma$ of sizes $(D+1) \times (D+1)$ and $D \times D$ respectively, and $y_k^\beta$, $\lambda_g^a \in \mathbb{R}$.*
- **Objective.** *Maximize the objective*

$$Q - \sum_{a=1}^{m} g^a \lambda_g^a + \int_0^T dt \left( \sum_{\beta=1}^{n} r_\beta \lambda_W^\beta(t) - \sum_{\gamma=1}^{l} h_\gamma \lambda_G^\gamma(t) \right), \tag{90}$$

*where $\lambda_G^\gamma(t)$ and $\lambda_W^\beta(t)$ are given by (88), subject to the following constraints:*

*1. **Primal objective.** For $a = 1, \ldots, m$,*

$$Q^a + \sum_{\beta=1}^{n} W_\beta^a \sum_{k=0}^{2D} y_k^\beta T^k = 0. \tag{91}$$

*2. **Dual master equation.** For $a = 1, \ldots, m$,*

$$\sum_{\beta=1}^{n} \left( -\frac{d\lambda_W^\beta(t)}{dt} W_\beta^a + \lambda_W^\beta(t) V_\beta^a \right) - \sum_{\gamma=1}^{l} \lambda_G^\gamma(t) G_\gamma^a = 0, \quad \forall t \in \mathbb{R}, \tag{92}$$

*where $\lambda_G^\gamma(t)$ and $\lambda_W^\beta(t)$ are given by (88).*
*3. **Dual probability bound.** For $\gamma = 1, \ldots, l$,*

$$Y_1^\gamma \succeq 0, \qquad Y_2^\gamma \succeq 0. \tag{93}$$

**4. Dual initial condition.** *For $a = 1, \dots, m$,*

$$\lambda_g^a - \sum_{\beta=1}^{n} y_0^\beta W_\beta^a = 0\,. \tag{94}$$

*The maximum value of the objective obtained by $DO_{poly}(L; D)$ is denoted as $\langle q \rangle_{t=T}^{min, DO_{poly}(L;D)}$.*

Note that there are only finitely many equations for the variables in constraint 2. More concretely, we can write

$$\sum_{\beta=1}^{n} \left( -\frac{d\lambda_W^\beta(t)}{dt} W_\beta^a + \lambda_W^\beta(t) V_\beta^a \right) - \sum_{\gamma=1}^{l} \lambda_G^\gamma(t) G_\gamma^a = \sum_{k=0}^{2D} t^k U_k^a(Y_1^\gamma, Y_2^\gamma, y_k^\beta; W_\beta^a, V_\beta^a, G_\gamma^a; T)\,, \tag{95}$$

where $U_k^a$ depends linearly on the variables $Y_1^\gamma$, $Y_2^\gamma$, and $y_k^\beta$. Constraint 2 is then equivalent to the following linear constraints on the variables:

$$U_k^a(Y_1^\gamma, Y_2^\gamma, y_k^\beta; W_\beta^a, V_\beta^a, G_\gamma^a; T) = 0\,, \quad a = 1, \dots, m\,, \quad k = 0, \dots, 2D\,. \tag{96}$$

Any feasible solution to $DO_{poly}(L; D)$ is also feasible for $DO(L)$. Therefore,

$$\langle q \rangle_{t=T}^{min, DO_{poly}(L;D)} \le \langle q \rangle_{t=T}^{min, DO(L)} \le \langle q \rangle_{t=T}\,. \tag{97}$$

We use `SemidefiniteOptimization` function in *Mathematica* with `Method → "MOSEK"` to solve the SDP problem $DO_{poly}(L; D)$. Unlike the exact LP solvers, SDP solvers based on the interior-point methods, including MOSEK, are numerical solvers subject to rounding errors. Nonetheless, they provide highly efficient numerical methods to solve problems like $DO_{poly}(L; D)$ at high $L$ values, and the numerical solver performance was very stable for all the examples discussed below. For the rest of this section, we will fix $D = 3$.

## 3.5 Results for the contact process on $\mathbb{Z}$

In the contact process, the infection density $\rho_t$ is a non-increasing function of $t$ with the initial condition $\left\langle \prod_{i \in A} s_i \right\rangle_{t=0} = 1$, $\forall A \subset \mathbb{Z}$ i.e. full infection [4]. We take this initial condition for $\lambda = 1$ contact process on $\mathbb{Z}$, and use $DO_{poly}(L; D = 3)$ to study how $\rho_t$ decreases

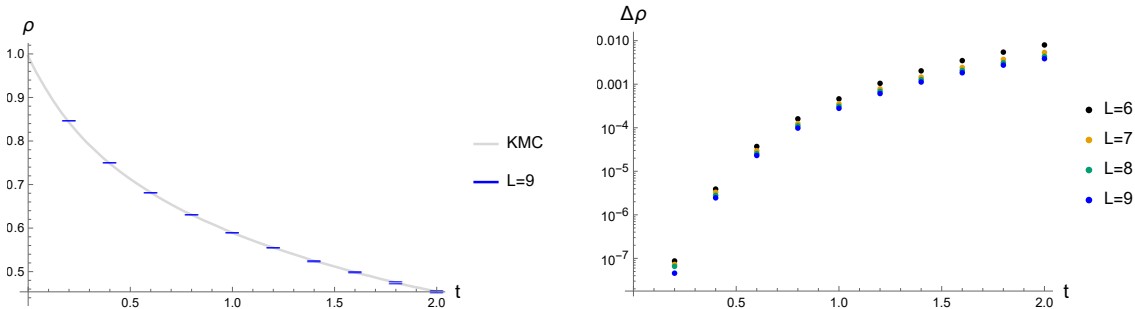

Figure 7: **Left**: $DO_{poly}(L = 9; D = 3)$ upper and lower bounds on $\rho$ as functions of time $t$ (blue) for the contact process on $\mathbb{Z}$ at $\lambda = 1$ with the initial condition $\left\langle \prod_{i \in A} s_i \right\rangle_{t=0} = 1$, $\forall A \subset \mathbb{Z}$, together with the KMC estimates obtained by averaging over 1000 independent simulations on a periodic lattice of size 200 (gray). Upper and lower bounds are hardly distinguishable. **Right**: Differences $\Delta\rho$ between $DO_{poly}(L; D = 3)$ upper and lower bounds on $\rho$ at $L = 6$ (black), $L = 7$ (orange), $L = 8$ (green), and $L = 9$ (blue).

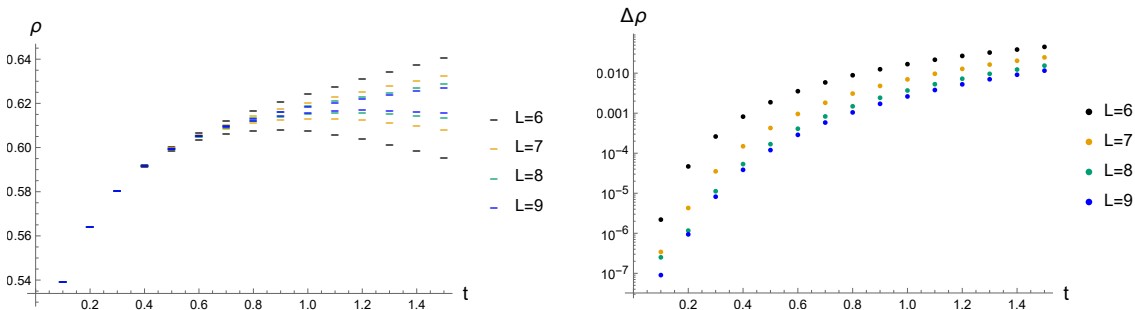

Figure 8: **Left**: $DO_{poly}(L; D = 3)$ upper and lower bounds on $\rho$ as functions of time $t$ for the contact process on $\mathbb{Z}$ at $\lambda = 2$ with the initial condition $\left\langle \prod_{i \in A} s_i \right\rangle_{t=0} = 0$, $\forall A \subset \mathbb{Z}$, for $L = 6$ (black), $L = 7$ (orange), $L = 8$ (green), and $L = 9$ (blue). **Right**: Differences $\Delta\rho$ between $DO_{poly}(L; D = 3)$ upper and lower bounds on $\rho$ at $L = 6$ (black), $L = 7$ (orange), $L = 8$ (green), and $L = 9$ (blue).

as $t$ grows. The resulting lower and upper bounds on $\rho_t$ are presented in Figure 7, along with the differences between them, which increase over time. In particular, we obtain from $DO_{poly}(L = 9; D = 3)$

$$0.5001 \leq \rho_{t=1.575}, \qquad \rho_{t=1.59} \leq 0.499991. \tag{98}$$

The half-life $t_{1/2}$ in the current example is defined by $\rho_{t=t_{1/2}} = \frac{1}{2}$. Therefore, we obtain lower and upper bounds (30) on $t_{1/2}$ at $\lambda = 1$.

With different initial conditions, $\rho_t$ does not need to be non-increasing. We apply $DO_{poly}(L; D = 3)$ to $\lambda = 2$ for the initial condition $\left\langle \prod_{i \in A} s_i \right\rangle_{t=0} = 0$, $\forall A \subset \mathbb{Z}$. In other words, we take the product of single site measures each of which has equal probabilities for $s_i = 1$ and $s_i = -1$. The results are presented in Figure 8. We note that at early times $t \lesssim 1.2$, $\rho_t$ increases and even passes the KMC estimate $\rho = 0.6036(6)$ for the upper invariant measure since $DO_{poly}(9; 3)$ produces a lower bound $0.617379 \leq \rho_{t=1.2}$. Therefore, we expect $\rho_t$ to eventually decrease, suggesting that $\rho_t$ is not monotonic in $t$. We do not have a *bootstrap proof* of such non-monotonicity from the presented results since the upper bound $\rho \leq 0.62267$ for the upper invariant measure obtained from $LP_{inv}(L = 10)$ at $\lambda = 2$ (Figure 1) is greater than the lower bounds on $\rho_t$ obtained from $DO_{poly}(9; 3)$ over $t \lesssim 1.5$. Nonetheless, the KMC estimate suggests that bounds obtained from higher values of $L$ will provide a proof that $\rho_t$ is non-monotonic in $t$.

## 3.6 Results for the asynchronous Domany-Kinzel model on $\mathbb{Z}$

We now consider the asynchronous Domany-Kinzel model on $\mathbb{Z}$ with $p_2 = 0$, which is non-monotonic. We begin by presenting a simple argument why $\rho_t$ monotonically decreases for $p_1 \leq \frac{1}{2}$ regardless of the initial conditions.

The master equation of $PO(L = 3)$ includes (assuming symmetries)

$$\frac{d}{dt}\langle s_1 \rangle_t = p_1 - 1 - \langle s_1 \rangle_t - p_1 \langle s_1 s_3 \rangle_t \quad \Rightarrow \quad \langle s_1 s_3 \rangle_t = -\frac{1}{p_1}\left(1 - p_1 + \langle s_1 \rangle_t + \frac{d}{dt}\langle s_1 \rangle_t\right). \tag{99}$$

The probability bound corresponding to the event $\{s \mid s_1 = 1, s_3 = 1\}$ then leads to

$$(2p_1 - 1)(1 + \langle s_1 \rangle_t) - \frac{d}{dt}\langle s_1 \rangle_t \geq 0, \tag{100}$$

implying that $\rho_t$ is a non-increasing function of $t$ for $p_1 \leq \frac{1}{2}$. We then define the half-life $t_{1/2}$ as usual: $\rho_{t=t_{1/2}} = \frac{1}{2}\rho_{t=0}$.

We apply $DO_{poly}(L = 6; 3)$ to the case $p_1 = \frac{1}{4}$ with the initial condition $\left\langle \prod_{i \in A} s_i \right\rangle_{t=0} = 1$, $\forall A \subset \mathbb{Z}$, and obtain

$$0.50024 \leq \rho_{t=0.81}, \qquad \rho_{t=0.811} \leq 0.499898, \tag{101}$$

leading to

$$0.81 < t_{1/2} < 0.811, \tag{102}$$

while the KMC estimate is given by $t_{1/2} \approx 0.8012(32)$.

## 3.7 Results for the contact process on $\mathbb{Z}^2$

It is straightforward to extend $DO_{poly}(L; D)$ to the analogous setup for processes on higher-dimensional lattices. We consider the contact process on $\mathbb{Z}^2$ in this section. Using the notations introduced in Definition 2 for $LP_{inv}^{2d}(L)$ and applying them to Definition 5, we obtain

**Definition 10.** *Given the objective function* $q(s) \in P_L'$, *initial conditions* $\left\langle \prod_{i \in A} s_i \right\rangle_{t=0} = y_A$ *where* $A \in \bar{D}_L^j$ *for* $j \in \partial \tilde{D}_{L+1}$, *respecting the lattice symmetries, and time* $T > 0$, $PO_2(L)$ *is a primal optimization problem where*

- **Variables.** *Variables are class* $C^1$ *functions* $\left\langle \prod_{i \in A} s_i \right\rangle_t$ *of* $t \in [0, T]$, *where* $A \subset D_L^j$ *for* $j \in \partial \tilde{D}_{L+1}$.
- **Objective.** *Minimize the objective* $\langle q(s) \rangle_{t=T}$ *subject to the following constraints:*

**1. Linearity.** *Given any polynomials* $q_1 \in P_L'$ *and* $q_2 \in P_L'$, *with* $\alpha \in \mathbb{R}$, *their expectation values satisfy linearity:* $\langle q_1 + \alpha q_2 \rangle_t = \langle q_1 \rangle_t + \alpha \langle q_2 \rangle_t$ *for* $t \in [0, T]$.
**2. Unit normalization.** $\langle 1 \rangle_t = 1$ *for* $t \in [0, T]$.
**3. Symmetry.** *For any* $A \subset \bar{D}_L^j$ *and* $B \subset \bar{D}_L^k$ *for* $j, k \in \partial \tilde{D}_{L+1}$ *such that* $A \sim B$, $\left\langle \prod_{i \in A} s_i \right\rangle_t = \left\langle \prod_{i \in B} s_i \right\rangle_t$ *for* $t \in [0, T]$.
**4. Master equation.** *For any polynomial* $f(s) \in \tilde{P}_L$,

$$\frac{d}{dt}\langle f(s) \rangle_t = \sum_{i \in \tilde{D}_L} \left\langle c(i, s)\left(f(\vec{s}^i) - f(s)\right) \right\rangle_t, \quad t \in [0, T], \tag{103}$$

*where* $c(i, s)$ *is given by* (5) *with* $d = 2$.
**5. Probability bound.** *For each* $j \in \partial \tilde{D}_{L+1}$, *and any given spin assignment* $u \in \{1, -1\}^{D_L^j}$,

$$\left\langle \prod_{i \in D_L^j} \frac{1 + u_i s_i}{2} \right\rangle_t \geq 0, \quad t \in [0, T]. \tag{104}$$

**6. Initial condition.** $\left\langle \prod_{i \in A} s_i \right\rangle_{t=0} = y_A$ *for* $A \in D_L^j$, $j \in \partial \tilde{D}_{L+1}$.

We can repeat the procedures described in sections 3.2 and 3.4 to formulate the dual optimization problem $DO_{poly}^2(L; D)$ of the primal optimization problem $PO_2(L)$ using the polynomial basis and obtain lower and upper bounds on time-dependent expectation values of interest. $DO_{poly}^2(L; D)$ takes exactly the same form as $DO_{poly}(L; D)$ except that inputs derived from the definition of the primal problems are different.

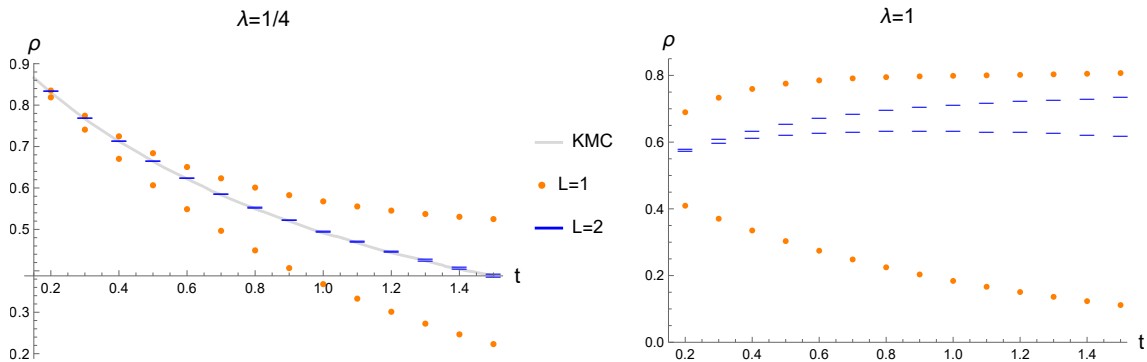

Figure 9: **Left:** $DO^2_{poly}(L; D = 3)$ upper and lower bounds on $\rho$ as functions of time $t$ for the contact process on $\mathbb{Z}^2$ at $\lambda = \frac{1}{4}$ with the initial condition $\left\langle \prod_{i \in A} s_i \right\rangle_{t=0} = 1$, $\forall A \subset \mathbb{Z}^2$, for $L = 1$ (orange) and $L = 2$ (blue), together with the KMC estimates obtained by averaging over 1000 independent simulations on $15 \times 15$ lattice (gray). **Right:** $DO^2_{poly}(L; D = 3)$ upper and lower bounds on $\rho$ as functions of time $t$ for the contact process on $\mathbb{Z}^2$ at $\lambda = 1$ with the initial condition $\left\langle \prod_{i \in A} s_i \right\rangle_{t=0} = 0$, $\forall A \subset \mathbb{Z}^2$, for $L = 1$ (orange) and $L = 2$ (blue).

Results at $\lambda = \frac{1}{4}$ (subcritical) and $\lambda = 1$ are presented in Figure 9. For $\lambda = \frac{1}{4}$, we take the initial condition $\left\langle \prod_{i \in A} s_i \right\rangle_{t=0} = 1$, $\forall A \subset \mathbb{Z}^2$. In particular, we obtain

$$0.50012 \leq \rho_{t=0.979}, \qquad \rho_{t=0.985} \leq 0.4998, \tag{105}$$

leading to the bounds $0.979 < t_{1/2} < 0.985$ on the half-life. This is consistent with the KMC estimate $t_{1/2} \approx 0.976(5)$. For $\lambda = 1$, the initial condition was taken to be $\left\langle \prod_{i \in A} s_i \right\rangle_{t=0} = 0$, $\forall A \subset \mathbb{Z}^2$.

## 3.8 Direct time evolution of the synchronous Domany-Kinzel model on $\mathbb{Z}$

As explained in the beginning of this section, time evolution of local expectation values in the synchronous Domany-Kinzel model is completely determined by (9) once initial conditions are given. Even though this is not a problem we solve using the bootstrap, we nonetheless discuss the results.

Consider the $p_2 = 0$ case and take the initial condition $\left\langle \prod_{i \in A} s_i \right\rangle_{t=1} = 0$, $\forall A \subset \mathbb{Z}$. From (10), we obtain

$$\langle s_i \rangle_{t=2} = p_1 - 1. \tag{106}$$

By repeating a similar exercise to obtain $C_A(f)$ for larger subsets $A \subset \mathbb{Z}$ and functions $f(s)$ depending on spins over larger subsets (assuming lattice symmetries), we can directly compute $\langle s_i \rangle_t$ at higher $t$:

$$\begin{aligned}
&\langle s_i \rangle_{t=3} = -p_1^3 + 2p_1^2 - 1, \quad \langle s_i \rangle_{t=4} = p_1^5 - 4p_1^4 + 4p_1^3 - 1, \\
&\langle s_i \rangle_{t=5} = -p_1^9 + 4p_1^8 - 6p_1^7 + 8p_1^6 - 12p_1^5 + 8p_1^4 - 1, \\
&\langle s_i \rangle_{t=6} = -p_1^{13} + 8p_1^{12} - 21p_1^{11} + 18p_1^{10} + 9p_1^9 - 28p_1^8 + 32p_1^7 - 32p_1^6 + 16p_1^5 - 1, \ \dots
\end{aligned} \tag{107}$$

Results up to $t = 9$ are plotted in Figure 10.

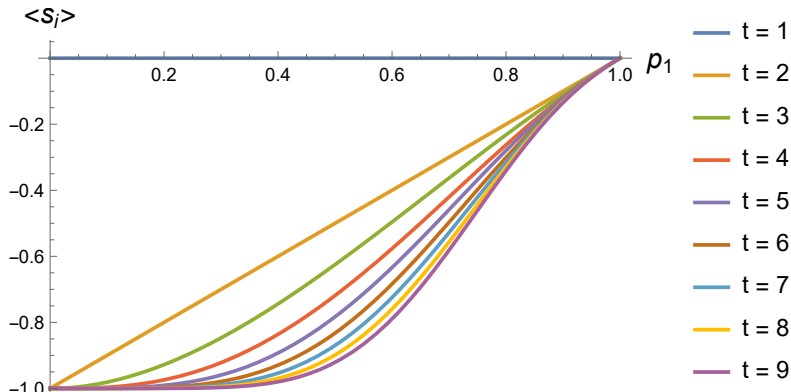

Figure 10: Time evolution of $\langle s_i \rangle$ for the synchronous Domany-Kinzel model on $\mathbb{Z}$ at $p_2 = 0$, as a function of $p_1$.

# 4  Bootstrapping the late-time evolution of noninvariant measures

Bootstrap constraints coming from the master equation and the probability bounds should also be obeyed at very late times. In the subcritical phase, the late-time behavior is governed by the temporal correlation length $\xi$, where expectation values approach those of the absorbing state exponentially as $\sim e^{-\frac{t}{\xi}}$. We now discuss how such decay behavior can be combined with the bootstrap constraints to derive bounds on $\xi$. These bounds are equivalent to bounds on the spectral gap $\Delta = \xi^{-1}$ of the time-evolution generator.

## 4.1  $\xi$ for the contact process on $\mathbb{Z}$

To be concrete, we consider the contact process on $\mathbb{Z}$. In the subcritical phase, for any initial condition satisfying $\langle s_i \rangle_{t=0} > -1$, we have:

$$\text{As } t \to \infty, \quad \left\langle \prod_{i \in A} s_i \right\rangle_t \to (-1)^{|A|} + B_A e^{-\frac{t}{\xi}}, \quad \forall A \subset \mathbb{Z}, \tag{108}$$

with $(-1)^{|A|} B_A < 0$. While the specific values of $B_A$ depend on the initial condition, the temporal correlation length $\xi$ is universal.

### 4.1.1  Analytic results at low $L$

Now consider the following master equation for the contact process involving spin variables over $D_{L=2}$ (assuming translation invariance):

$$\frac{d}{dt} \langle s_1 \rangle_t = -\lambda \langle s_1 s_2 \rangle_t - \langle s_1 \rangle_t + \lambda - 1. \tag{109}$$

By substituting (108) into (109), we obtain

$$-\frac{1}{\xi} B_{\{1\}} e^{-\frac{t}{\xi}} = -\lambda B_{\{1,2\}} e^{-\frac{t}{\xi}} - B_{\{1\}} e^{-\frac{t}{\xi}} \quad \Rightarrow \quad B_{\{1,2\}} = \frac{B_{\{1\}}}{\lambda} \left( \frac{1}{\xi} - 1 \right). \tag{110}$$

As expected, only the terms proportional to $e^{-\frac{t}{\xi}}$ remain, since the absorbing state is invariant.

For the probability bounds, we know that $\left\langle \prod_{i \in A} \frac{1+u_i s_i}{2} \right\rangle_{t \to \infty} = 1 \geq 0$ when $u_i = -1$ for all $i \in A$. In contrast, when there exists $i \in A$ such that $u_i = 1$, the quantity $\left\langle \prod_{i \in A} \frac{1+u_i s_i}{2} \right\rangle_{t \to \infty}$

decays exponentially to 0 as $\sim e^{-\frac{t}{\xi}}$, potentially leading to nontrivial constraints on the coefficients of the exponential terms:

$$\lim_{t \to \infty} \left\langle \prod_{i \in A} \frac{1 + u_i s_i}{2} \right\rangle_t = \mathcal{M}_u e^{-\frac{t}{\xi}} \geq 0 \text{ if } \exists i \in A \text{ s.t. } u_i = 1 \quad \Rightarrow \quad \mathcal{M}_u \geq 0, \tag{111}$$

where $\mathcal{M}_u$ is independent of $t$.

For example, the probability bounds for the events $\{s \mid s_1 = 1, s_2 = 1\}$ and $\{s \mid s_1 = 1, s_2 = -1\}$ as $t \to \infty$ lead to

$$0 \leq 2B_{\{1\}} + B_{\{1,2\}} = B_{\{1\}}\left(2 + \frac{1}{\lambda}\left(\frac{1}{\xi} - 1\right)\right), \qquad 0 \leq -B_{\{1,2\}} = -\frac{B_{\{1\}}}{\lambda}\left(\frac{1}{\xi} - 1\right). \tag{112}$$

Together with $B_{\{1\}} > 0$ and $B_{\{1,2\}} < 0$, we conclude

$$1 < \xi \leq \frac{1}{1 - 2\lambda}, \quad \text{if } \lambda < \frac{1}{2}, \qquad 1 < \xi, \quad \text{if } \frac{1}{2} \leq \lambda < \lambda_c. \tag{113}$$

Note that the upper bound on $\xi$ diverges at $\lambda = \frac{1}{2}$, which coincides with the lower bound on the critical rate $\lambda_c$ obtained from $LP_{inv}(L = 2)$. This correspondence persists at higher $L$. In section 2, we used $LP_{inv}(L)$ to derive a lower bound $\lambda_L$ on $\lambda_c$ at each $L$. For $\lambda < \lambda_L$, the solution to the invariance equation in $LP_{inv}(L)$ is uniquely given by the absorbing state. Since the late-time master equation with $\xi = \infty$ reduces to the invariance equation, we would obtain $B_A = 0$ for $\lambda < \lambda_L$ if $\xi = \infty$, contradicting the requirement $(-1)^{|A|}B_A < 0$. Therefore, $\xi = \infty$ is allowed only if $\lambda \geq \lambda_L$.

We now extend the analysis to higher $L$, i.e., we consider expectation values of spin variables over $D_L$ for $L = 3, 4, \ldots$. Starting from $L = 3$, the probability bounds $\left\langle \prod_{i \in D_L} \frac{1 + u_i s_i}{2} \right\rangle_{t \to \infty} \geq 0$ generally depend on multiple $B_A$'s for $A \subset D_L$, even after applying symmetries and the master equation. For example, at $L = 3$, we can eliminate $B_{\{1,2\}}$ and $B_{\{1,2,3\}}$ using the master equation, but two independent variables remain: $B_{\{1\}}$ and $B_{\{1,3\}}$. After imposing symmetries and the master equation, the probability bounds reduce to

$$\begin{aligned}
(1 + 2\xi^2 - \xi(3 + 2\lambda))B_{\{1\}} &\geq 0, \\
2\lambda^2\xi^2 B_{\{1,3\}} + \left(2(\lambda + 1)\xi^2 - (4\lambda + 3)\xi + 1\right)B_{\{1\}} &\geq 0, \\
\left((2\lambda + 3)\xi - 2\xi^2 - 1\right)B_{\{1\}} - 2\lambda^2\xi^2 B_{\{1,3\}} &\geq 0, \\
\left(2(\lambda - 1)\xi^2 + 3\xi - 1\right)B_{\{1\}} &\geq 0, \\
2\lambda^2\xi^2 B_{\{1,3\}} + \left(\left(4\lambda^2 - 2\lambda + 2\right)\xi^2 - 3\xi + 1\right)B_{\{1\}} &\geq 0.
\end{aligned} \tag{114}$$

We can form positive linear combinations of these inequalities to eliminate $B_{\{1,3\}}$, obtaining inequalities that depend only on $B_{\{1\}}$:

$$\begin{aligned}
(1 + 2\xi^2 - \xi(3 + 2\lambda))B_{\{1\}} &\geq 0, \\
\lambda(\xi - 1)\xi B_{\{1\}} &\geq 0, \\
\left(2(\lambda - 1)\xi^2 + 3\xi - 1\right)B_{\{1\}} &\geq 0, \\
((2\lambda - 1)\xi + 1)B_{\{1\}} &\geq 0.
\end{aligned} \tag{115}$$

Together with $B_{\{1\}} > 0$, we conclude

$$\frac{1}{4}\sqrt{4\lambda^2 + 12\lambda + 1} + \frac{1}{4}(2\lambda + 3) \leq \xi \leq \frac{1}{4}\sqrt{\frac{8\lambda + 1}{(\lambda - 1)^2} - \frac{3}{4(\lambda - 1)}}, \quad \text{if } \lambda < 1. \tag{116}$$

The analogous elimination at $L = 4$ is still tractable analytically. It results in

$$x^*_{4,min} \leq \xi \leq x^*_{4,max}, \quad \text{if } \lambda < \frac{1 + \sqrt{37}}{6}, \tag{117}$$

where $x^*_{4,min}$ is the largest real root of the polynomial

$$6x^3 + x^2\left(-4\lambda^2 - 8\lambda - 11\right) + x(4\lambda + 6) - 1, \tag{118}$$

for $0 < \lambda < \frac{1+\sqrt{37}}{6}$, and $x^*_{4,max}$ is the largest real root of the polynomial

$$x^4\left(12\lambda^2 - 4\lambda - 12\right) + x^3(22\lambda + 28) + x^2\left(-4\lambda^2 - 18\lambda - 23\right) + x(4\lambda + 8) - 1, \tag{119}$$

for the same range of $\lambda$.

### 4.1.2 LP results at higher $L$

Combining probability bounds to eliminate all but $B_{\{1\}}$ is in principle possible at higher $L$, but in practice computationally expensive. Instead, we reformulate the problem as a LP feasibility question. At a fixed trial value $\xi = \xi^*$, we ask whether there exist coefficients $B_A$ such that the probability bounds and the master equation, together with the condition $(-1)^{|A|}B_A < 0$, are satisfied. If no such $B_A$ exists, then the actual value of $\xi$ cannot be $\xi^*$. By scanning over trial values of $\xi^*$, we can determine which values of $\xi$ are allowed and which are excluded.

In the low-$L$ analysis above, we observed that the set of allowed $\xi$ values forms a single interval. This is natural, as the constraints from the probability bounds and the master equation imply that late-time expectation values cannot decay arbitrarily fast or arbitrarily slowly, while all intermediate decay rates are likely to be admissible. We will assume that this single-interval structure persists at higher $L$. While this assumption could be investigated directly, we do not pursue that direction here. Explicit feasibility checks at finely spaced trial values of $\xi$ provide strong empirical support for the assumption.

Under the single-interval assumption, we extract upper and lower bounds on $\xi$ as follows. First, we identify a trial value $\xi = \xi_0$ for which the constraints are feasible. We then increase $\xi$ gradually from $\xi_0$ until feasibility fails, at which point we obtain an upper bound on $\xi$. Similarly, decreasing $\xi$ from $\xi_0$ until feasibility fails yields a lower bound. Feasibility tests can be carried out using the following LP:

**Definition 11.** *Given a trial value $\xi = \xi^* > 0$, $LP_{\xi*}(L)$ is a LP problem where*
- **Variables.** *Variables are $B_A$, where $A \subset D_L$.*
- **Objective.** *Maximize the objective $B_{\{1\}}$ subject to the following constraints:*

*1. late-time behavior. Make substitutions*

$$\left\langle \prod_{i\in A} s_i \right\rangle_t \to (-1)^{|A|} + B_A e^{-\frac{t}{\xi^*}}, \quad \forall A \subset D_L, \tag{120}$$

*for all the expressions $\left\langle \prod_{i\in A} s_i \right\rangle_t$ appearing below.*
*2. Linearity. Given any polynomials $q_1 \in P_L$ and $q_2 \in P_L$, with $\alpha \in \mathbb{R}$, their expectation values satisfy linearity: $\langle q_1 + \alpha q_2 \rangle_t = \langle q_1 \rangle_t + \alpha \langle q_2 \rangle_t$.*
*3. Unit normalization. $\langle 1 \rangle_t = 1$.*
*4. Symmetry. For any $A \subset D_L$ and $A' \subset D_L$ such that $A \sim A'$, $B_A = B_{A'}$.*
*5. Master equation. For any polynomial $f(s) \in P_{L-1}$,*

$$\frac{d}{dt}\langle f(s) \rangle_t = \sum_{i\in D_{L-1}} \left\langle c(i,s)\left(f(\bar{s}^i) - f(s)\right) \right\rangle_t, \tag{121}$$

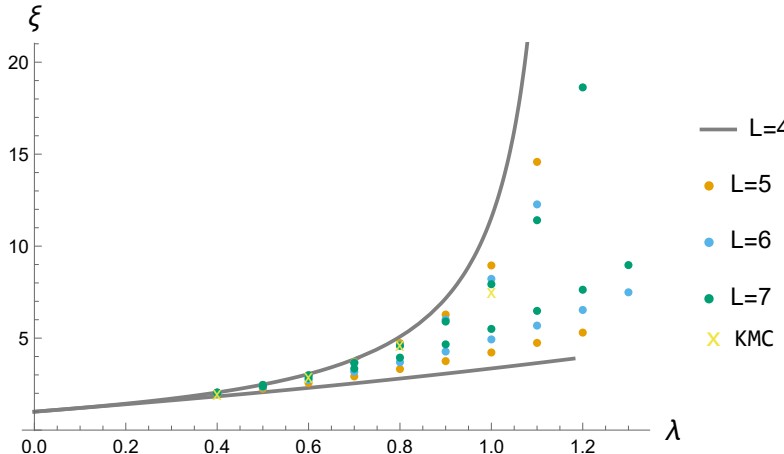

Figure 11: Upper and lower bounds on $\xi$ obtained from $LP_{\xi^*}(L)$ at $L = 5$ (orange), $L = 6$ (blue), and $L = 7$ (green), together with the analytic bounds (117) at $L = 4$ (gray) and KMC estimates (yellow). Some finite but large upper bounds lie outside of the scale of the current plot.

*where* $\left\langle \prod_{i \in A} s_i \right\rangle_t$ *with* $0 \in A$ *is replaced by* $\left\langle \prod_{i \in \tau_+(A)} s_i \right\rangle_t$ *so that the equation closes within the variables under consideration.*

**6. Probability bound.** *For any given spin assignment* $u \in \{1, -1\}^{D_L}$ *except for* $u_i = -1$, $\forall i \in D_L$,

$$\left\langle \prod_{i \in D_L} \frac{1 + u_i s_i}{2} \right\rangle_t \geq 0 \,. \tag{122}$$

**7. Boundedness.** $B_{\{1\}} \leq 1$.

A few remarks are in order. First, the master equation is homogeneous in $B_A$ since the absorbing state is invariant. Likewise, the probability bounds considered above are homogeneous in $B_A$, as the events they correspond to involve at least one spin variable being $+1$, whose probability decays to zero at late times. Therefore, $B_A = 0$ is always a feasible solution to $LP_{\xi^*}(L)$. If there exists a feasible solution with $B_{\{1\}} \neq 0$, then any overall positive rescaling of the $B_A$'s also yields a feasible solution for all constraints, possibly except for the boundedness condition $B_{\{1\}} \leq 1$. Thus, $LP_{\xi^*}(L)$ has only two possible outcomes: $B_{\{1\}} = 0$ or $B_{\{1\}} = 1$.

When the outcome is $B_{\{1\}} = 0$, it implies that the only feasible value of $B_{\{1\}}$ is zero. This contradicts the condition $(-1)^{|A|} B_A < 0$ and therefore excludes $\xi^*$ from the set of allowed values of $\xi$. In contrast, when the outcome is $B_{\{1\}} = 1$, we then check whether $\prod_{A \subset D_L} B_A \neq 0$ holds for the solution of $LP_{\xi^*}(L)$. If so, the condition $(-1)^{|A|} B_A < 0$ follows from the probability bounds, and $\xi^*$ is accepted as an allowed value of $\xi$.

We applied $LP_{\xi^*}(L)$ at $L = 5, 6, 7$ over finely spaced values of $\xi^*$ using the `LinearOptimization` function in *Mathematica*, and obtained the upper and lower bounds on $\xi$ shown in Figure 11, under the single-interval assumption discussed above. Rough KMC estimates of $\xi$ were obtained as described in Appendix B.

## 4.2  $\xi$ for the contact process on $\mathbb{Z}^2$

It is straightforward to extend the late-time analysis to higher dimensions. Here, we consider the contact process on $\mathbb{Z}^2$. Using the notations from section 2.4, we introduce the following LP problem:

**Definition 12.** *Given a trial value $\xi = \xi^* > 0$, $LP^{2d}_{\xi^*}(L)$ is a LP problem where*

- **Variables.** *Variables are $B_A$, where $A \subset \bar{D}^j_L$ for $j \in \partial \tilde{D}_{L+1}$.*
- **Objective.** *Maximize the objective $B_{\{(0,0)\}}$ subject to the following constraints:*

**1. Late-time behavior.** *Make substitutions*

$$\left\langle \prod_{i \in A} s_i \right\rangle_t \to (-1)^{|A|} + B_A e^{-\frac{t}{\xi^*}}, \quad \forall A \subset \bar{D}^j_L, \quad j \in \partial \tilde{D}_{L+1}, \tag{123}$$

*for all the expressions $\left\langle \prod_{i \in A} s_i \right\rangle_t$ appearing below.*

**2. Linearity.** *Given any polynomials $q_1 \in P'_L$ and $q_2 \in P'_L$, with $\alpha \in \mathbb{R}$, their expectation values satisfy linearity: $\langle q_1 + \alpha q_2 \rangle_t = \langle q_1 \rangle_t + \alpha \langle q_2 \rangle_t$.*

**3. Unit normalization.** $\langle 1 \rangle_t = 1$.

**4. Symmetry.** *For any $A \subset \bar{D}^j_L$ and $A' \subset \bar{D}^k_L$ for $j, k \in \partial \tilde{D}_{L+1}$ such that $A \sim A'$, $B_A = B_{A'}$.*

**5. Master equation.** *For any polynomial $f(s) \in \tilde{P}_L$,*

$$\frac{d}{dt} \langle f(s) \rangle_t = \sum_{i \in \tilde{D}_L} \left\langle c(i,s) \left( f(\bar{s}^i) - f(s) \right) \right\rangle_t. \tag{124}$$

**6. Probability bound.** *For each $j \in \partial \tilde{D}_{L+1}$, and any given spin assignment $u \in \{1, -1\}^{\bar{D}^j_L}$ except for $u_i = -1$, $\forall i \in \bar{D}^j_L$,*

$$\left\langle \prod_{i \in \bar{D}^j_L} \frac{1 + u_i s_i}{2} \right\rangle_t \geq 0. \tag{125}$$

**7. Boundedness.** $B_{\{(0,0)\}} \leq 1$.

Again, only two outcomes are possible: $B_{\{(0,0)\}} = 0$ or $B_{\{(0,0)\}} = 1$. If $B_{\{(0,0)\}} = 0$, then $\xi^*$ is excluded as a valid value of $\xi$. At $L = 1$, the analysis can be carried out analytically. The master equation, together with symmetries, leads to

$$-\frac{1}{\xi} B_{\{(0,0)\}} = B_{\{(0,0)\}} - 2\lambda B_{\{(0,0),(1,0)\}}. \tag{126}$$

The probability bounds are then given by

$$(\xi - 1) B_{\{(0,0)\}} \geq 0, \qquad ((4\lambda - 1)\xi + 1) B_{\{(0,0)\}} \geq 0, \tag{127}$$

which imply

$$1 < \xi \leq \frac{1}{1 - 4\lambda}, \quad \text{if } \lambda < \frac{1}{4}, \qquad 1 < \xi, \quad \text{if } \frac{1}{4} \leq \lambda < \lambda_c. \tag{128}$$

At $L = 2$, we perform the feasibility test described in the previous section and obtain

$$1.48 \leq \xi \leq 1.528 \ \text{ at } \lambda = 0.1, \qquad 2.126 \leq \xi \leq 2.665 \ \text{ at } \lambda = 0.2, \tag{129}$$

while rough KMC estimates are given by $\xi \approx 1.411$ at $\lambda = 0.1$ and $\xi \approx 2.489$ at $\lambda = 0.2$ (see Appendix B).

# 5 Future prospects

In this work, we presented bootstrap methods implementing the defining properties—such as positivity, invariance, or the master equation—of the measures of interest in nonequilibrium

stochastic processes, and derived nontrivial bounds on expectation values and other relevant quantities. It is guaranteed that the bounds will eventually converge to the actual values as more and more constraints are imposed [9]. There are several clear directions for future investigation.

- It is desirable to increase the level $L$ of the optimization problems to obtain tighter bounds. Although the size of the problems grows exponentially with $L$, it is plausible that only a small subset of the constraints plays a significant role in improving the bounds. In the context of quantum mechanical systems on a lattice, identifying the most *relevant* subset of bootstrap constraints among an exponentially large set has been recently studied and shown to significantly improve the resulting bounds [36, 51]. This identification relies on tensor network approaches to quantum many-body systems, which exploit the entanglement structure of low-energy states. Tensor network methods have also been applied to nonequilibrium stochastic processes in earlier works [52, 53], and it would be interesting to explore whether they can be used to construct more efficient bootstrap methods and thereby yield improved bounds.

- Even though we derived lower bounds on the critical rates, the current bootstrap methods for invariant measures do not provide a mechanism to obtain upper bounds. Such upper bounds have been derived previously using alternative methods (see, e.g., [7] for the case of the contact process on $\mathbb{Z}$), but it remains unclear whether approaches based solely on the positivity of the measures can yield similar results.

In fact, the derivation of even a nontrivial one-sided bound on the critical rate from the positivity of measures is not guaranteed in general stochastic processes. In the equilibrium stochastic Ising model, bootstrap methods for invariant or reversible measures did not yield bounds on the critical temperature unless additional ingredients—such as the first Griffiths inequality [54, 55]—were employed [33]. In the nonequilibrium setting, bootstrap methods for the invariant measures of the asynchronous version of Toom's rule [56] do not appear to produce bounds on the critical rates unless the bias is maximal [57].

We expect that by systematically investigating the probability bounds involving one spin and two adjacent spins for all nearest-neighbor transition rules with symmetries, one may uncover general criteria for when bootstrap methods can constrain the critical rate. A related question—suggested to have an affirmative answer by the results of this work—is whether bootstrap methods always yield at least one-sided bounds on the critical rates for processes with absorbing states.

- In section 4, we derived bounds on the temporal correlation length $\xi$ in the subcritical phase. At criticality, we expect power-law behavior of the expectation values at late times. It would be extremely interesting if bootstrap methods could provide nontrivial bounds on the associated critical exponents—either through the approach discussed in this work or via methods more closely aligned with the conformal bootstrap [37–39, 58].

The absence of conformal symmetry, reflection positivity, time-reversal symmetry, and Hermiticity of the time-evolution generator introduces unique challenges in studying critical nonequilibrium processes such as DP criticality. Nonetheless, scale invariance is still present and may prove useful in gaining a deeper understanding of these systems. Recent developments in equilibrium scale-invariant theories that lack conformal invariance and reflection positivity [59] may also offer valuable insights for advancing the bootstrap program in nonequilibrium settings.

- The study of late-time behavior in section 4 was enabled by the presence of an absorbing state. In particular, deviation terms with exponential decay contributed nontrivially to the probability bounds precisely because most of these probabilities are identically zero in the absorbing state. It is natural to ask whether a similar analysis can be applied to processes without an absorbing state.

Even in cases where there is a unique invariant measure and all its expectation values are explicitly known, if the measure is generic in the sense that typical probabilities are nonzero, the probability bounds will be trivially satisfied at late times. Nonetheless, it may still be possible that additional properties of the initial conditions or of the dynamics—such as certain monotonicity—lead to constraints on the coefficients of the exponentially decaying terms. When combined with the master equation, such constraints may yield nontrivial results.

• The bootstrap methods discussed in this work have direct applications to problems in other fields of mathematical sciences. For classical dynamical systems [19], it is straightforward to bound the time evolution of non-stationary measures. An interesting question is whether such studies might shed light on the nature of strange attractors in chaotic systems such as the Lorenz system, whose properties have been difficult to establish using bootstrap methods based on stationary measures. Similarly, in the context of quantum mechanical systems [13], there are many compelling questions concerning the late-time behavior of correlators that would be fascinating to explore. The main challenge, once again, appears to lie in identifying inequality constraints that remain nontrivial at late times.

## Acknowledgments

We would like to thank Barak Gabai, David Goluskin, Jong Yeon Lee, Henry Lin, Yuan Xin, and Zechuan Zheng for discussions. We are especially grateful to Yuan Xin for suggesting the bootstrap analysis of the ratios $\mathcal{R}$ for upper invariant measures.

**Funding information**  This work is supported by Clay Córdova's Sloan Research Fellowship from the Sloan Foundation.

## A   Size and runtime of the optimization problems

In this section, we present a few details about the optimization problems employed in this work.

### A.1   LP for invariant measures

When implementing $LP_{inv}(L)$ in Definition 1 using the LinearOptimization function in *Mathematica*, we explicitly solved the symmetry constraint 3 and the invariance constraint 4 using the linearity constraint 1 and the unit normalization constraint 2, in order to reduce the problem to the minimal number of independent variables. We also applied the symmetry constraint 3 to the probability bound constraint 5 to reduce the number of inequalities in the LP. The number of variables and probability bounds at different values of $L$ is presented in Table 2 for the case of the contact process on $\mathbb{Z}$.

While solving the linear equality constraints required negligible time (less than a minute for $L = 10$), the runtime for the LinearOptimization solver increased significantly with $L$. For a single data point measured on a laptop, the runtime at different values of $L$ is summarized in Table 3.

For the contact process on $\mathbb{Z}^2$, $LP_{inv}^{2d}(L = 2)$ in Definition 2 has 50 variables after imposing symmetry constraints, 40 variables after further imposing invariance constraints, and 272 probability bounds. The runtime of LinearOptimization in this case was approximately 30 seconds.

Table 2: Number of independent variables in $LP_{inv}(L)$ for the contact process on $\mathbb{Z}$ after imposing symmetry and invariance constraints, along with the number of probability bounds. Without any constraints, the number of variables is $2^L - 1$.

| $L$ | 2 | 3 | 4 | 5 | 6 | 7 | 8 | 9 | 10 |
|---|---|---|---|---|---|---|---|---|---|
| # of variables after symmetries | 2 | 4 | 7 | 13 | 23 | 43 | 79 | 151 | 287 |
| # of variables after symmetries and invariance | 1 | 2 | 3 | 6 | 10 | 20 | 36 | 72 | 136 |
| # of probability bounds | 3 | 6 | 10 | 20 | 36 | 72 | 136 | 272 | 528 |

Table 3: `LinearOptimization` solver runtime for $LP_{inv}(L)$ for the contact process on $\mathbb{Z}$. For $L < 7$, the runtime was negligible.

| $L$ | 7 | 8 | 9 | 10 |
|---|---|---|---|---|
| Runtime | a few seconds | $\sim 30$ seconds | $\sim 5$ minutes | $\sim 3$ hours |

### A.2 SDP for noninvariant measures

For the time evolution of noninvariant measures in asynchronous stochastic processes, we solve the dual optimization problems. We focus on $DO_{poly}(L; D = 3)$ from Definition 9, which employs a polynomial basis of degree up to 6. As described in section 3.2, we solve for the symmetry constraints in the primal optimization problem in advance, which determines the values of $m$, $n$, and $l$, and thereby fixes the size of $DO_{poly}(L; D = 3)$. Specifically, $m$ is the number of primal variables after imposing symmetries (second row of Table 2); $n$ is the difference between the number of primal variables before and after imposing the invariance equations (i.e., the difference between the second and third rows of Table 2); and $l$ is the number of probability bounds (last row of Table 2).

Once $DO_{poly}(L; D = 3)$ is expressed as in Definition 9, we solve a total of $3m$ linear equality constraints explicitly to obtain the minimal number of independent variables, and then impose the semidefinite constraints using the `SemidefiniteOptimization` function in *Mathematica* with `Method → "MOSEK"` (with default parameter values). In total, there are $2l$ positive semidefinite matrices: $l$ of size $(D+1) \times (D+1)$ and another $l$ of size $D \times D$. Solving the linear equality constraints using `NSolve` (with default settings) often takes comparable—or even greater—time than the semidefinite optimization step itself. The runtimes for both procedures are summarized in Table 4 for the contact process on $\mathbb{Z}$.

For $DO^2_{poly}(L = 2; D = 3)$ applied to the contact process on $\mathbb{Z}^2$, the `NSolve` runtime was approximately one minute. The `SemidefiniteOptimization` runtime was also about one minute for the initial condition $\left\langle \prod_{i \subset A} s_i \right\rangle_{t=0} = 1$ in the subcritical phase, and about ten minutes for the initial condition $\left\langle \prod_{i \subset A} s_i \right\rangle_{t=0} = 0$ in the supercritical phase.

## B Kinetic Monte Carlo simulations

In this section, we briefly describe the KMC simulations used in this work. For a more comprehensive introduction to KMC, we refer the reader to [60]. KMC is employed to simulate asynchronous stochastic processes on a finite lattice $\Lambda \subset \mathbb{Z}^d$.

Table 4: Runtime for `NSolve` and `SemidefiniteOptimization` functions for $DO_{poly}(L; D = 3)$ in the case of the contact process on $\mathbb{Z}$.

| $L$ | 6 | 7 | 8 | 9 |
|---|---|---|---|---|
| `NSolve` runtime | ~1 second | ~10 seconds | ~3 minutes | ~10 minutes |
| `SemidefiniteOptimization` runtime | ~1 second | ~10 seconds | ~1 minute | ~1.5 minutes |

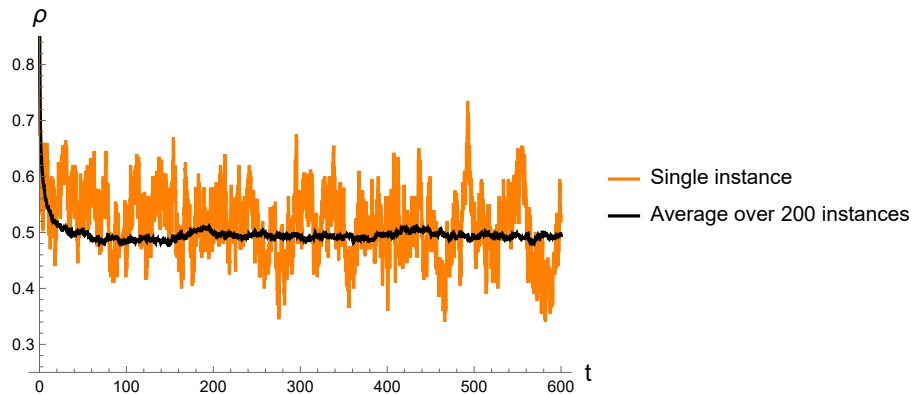

Figure 12: KMC simulation results for the infection density $\rho$ of the contact process on a periodic lattice of size 200 at $\lambda = 1.8$. The initial state is given by $s_i = 1$ for all $i \in \Lambda$. A single KMC instance is shown in orange, while the average over 200 simulations is shown in black.

Recall that in asynchronous processes, the current spin configuration $s$ can transition to a new configuration $\bar{s}^i$ spontaneously at a rate $c(i, s)$ for all $i \in \Lambda$. On a finite lattice, there are only finitely many possible transitions, and the total rate is given by $R_s = \sum_{i \in \Lambda} c(i, s)$. Consequently, the waiting time $t_s$ until the next transition (after the system has entered configuration $s$) is drawn from an exponential distribution with rate parameter $R_s$: $t_s \sim \text{Exp}(R_s)$.[14] Once $t_s$ is drawn, a site $i \in \Lambda$ is selected with probability $c(i, s)/R_s$, and the spin configuration is updated to $\bar{s}^i$. The KMC simulation procedure is summarized in Algorithm 1.

---
**Algorithm 1** KMC simulation of asynchronous stochastic processes up to time $T$
---
1: Start with an initial state $s$ at time $t = 0$.
2: **while** $t < T$ **do**
3:     Compute $R_s = \sum_{i \in \Lambda} c(i, s)$.
4:     Draw $t_s \sim \text{Exp}(R_s)$ and update $t = t + t_s$.
5:     Randomly choose a site $i \in \Lambda$ with probability $c(i, s)/R_s$.
6:     Update $s = \bar{s}^i$.
7: **end while**
---

In Figure 12, we present the KMC simulation results for the contact process on a periodic lattice of size 200 at $\lambda = 1.8$, with the initial state given by $s_i = 1$ for all $i \in \Lambda$. To obtain the KMC estimates for the invariant measures in section 2, we first computed the time average

---
[14]The exponential distribution describes the time intervals between consecutive events in a Poisson point process, which defines the notion of time (set by Poisson clocks) in asynchronous stochastic processes.

of $\rho$ for each simulation after discarding a transient period. These time averages were then averaged over 200 independent runs to produce the final KMC estimate for $\rho$ (and similarly for $\nu$). For the time-evolution results in section 3, KMC estimates were obtained by averaging over 1000 independent simulations.

KMC estimates for the critical rates can be inferred from the late-time power-law decay of $\rho$. For example, in the asynchronous Domany-Kinzel model with $p_2 = 0$, we expect $\rho_t \sim t^{-\delta}$ at the critical point $p_1 = p_{1c}$, where $\delta \approx 0.159464(6)$ is the universal critical exponent for critical DP in $1+1$ dimensions [61]. In Figure 13, we show log-log plots of $\rho$ vs. $t$ at $p_1 = 0.9$ (subcritical), $p_1 = 0.908$ (critical), and $p_1 = 0.93$ (supercritical), where a clear power-law decay emerges at late times $t \gtrsim 100$ for $p_1 = 0.908$. The fit yields $\delta \approx 0.15992$.

KMC estimates for the temporal correlation length $\xi$ in the subcritical phase are extracted from the exponential decay of $\rho$. Figure 14 presents an example for the contact process on a periodic lattice of size 200 at $\lambda = 0.8$, where the slope of the log plot gives an estimate of the spectral gap $\Delta = \xi^{-1}$. Other estimates of $\xi$ in section 4 were obtained using similar procedures, averaging over 1000 independent KMC simulations. For the contact process on $\mathbb{Z}^2$, simulations were performed on a periodic $15 \times 15$ lattice.

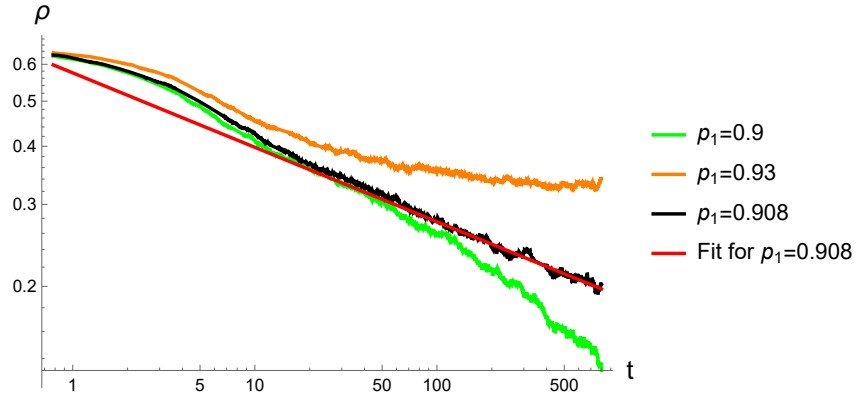

Figure 13: Log-log plot of $\rho$ vs. $t$ from KMC simulations of the asynchronous Domany-Kinzel model with $p_2 = 0$ on a periodic lattice of size 200, averaged over 200 independent runs, at $p_1 = 0.9$ (green), $p_1 = 0.908$ (black), and $p_1 = 0.93$ (orange). The initial state is given by $s_i = 1$ if $i \equiv 1,2 \mod 3$ and $s_i = -1$ if $i \equiv 0 \mod 3$. A power-law fit $\rho(t) = 0.57507\, t^{-0.15992}$ at $p_1 = 0.908$ obtained from $t \in [100, 640]$ is shown in red.

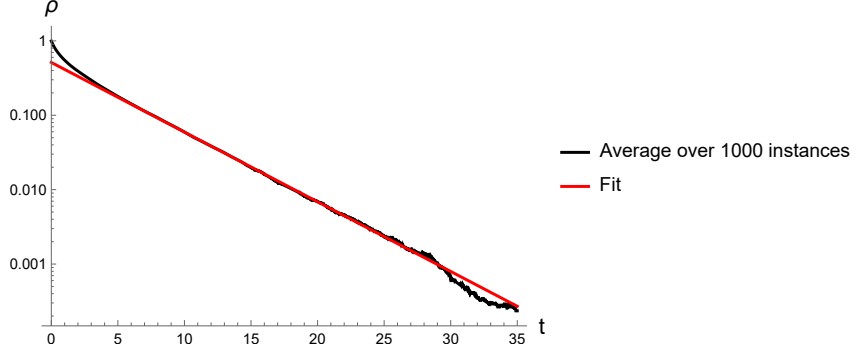

Figure 14: Log plot of $\rho$ vs. $t$ from KMC simulations of the contact process with $\lambda = 0.8$ on a periodic lattice of size 200, averaged over 1000 independent runs (black). The initial state is fully infected: $s_i = 1$ for all $i$ on the lattice. A fit $\rho(t) = 0.512977\, e^{-0.215633t}$ obtained from $t \in [5, 25]$ is shown in red.

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
