# Peer review of "Bootstrapping Nonequilibrium Stochastic Processes"

_SciPost Physics, doi:SciPost Phys. 19, 124 (2025)_

## Round 3 · Referee Report · Anonymous (Referee 2) · 2025-9-9

Strengths
1, The paper provides a novel bootstrap method for studying non-equilibrium stochastic processes. The method is based on general constraints. The author showed it works for both stationary and time-dependent cases, using LP and SDP respectively. The method has potential to apply to more problems beyond the examples studied in the paper.
2, The LP result is mathematically rigorous.
3, The text is well written.
Weaknesses
Report
The paper meets the scipost Acceptance Criteria Expectations 1 and 2 and satisfies all General acceptance criteria. Therefore I recommand it for publishing on scipost.
Recommendation
Publish (easily meets expectations and criteria for this Journal; among top 50%)
Report
Requested changes
- on page 18, an equation is coming off the page, and should be fixed
- The author gives many examples of bounds that are consistent with previous Monte Carlo methods. Could the author also discuss examples where previous methods are insufficient, and the bootstrap is the only way forward? Does the author already have results in that direction, or is that only a future direction?
Recommendation
Ask for minor revision

Author: Minjae Cho on 2025-08-19 [id 5740]
(in reply to Report 1 on 2025-08-14)First of all, thank you very much for referring my work and providing helpful comments.
I will fix the equation which came off the page in the next version.
One major relative advantage of bootstrap methods compared to Monte Carlo methods in general is that the former provides mathematically rigorous results, while the latter provides only estimates. Even for the results where bootstrap and Monte Carlo are consistent, this conceptual difference still persists. Furthermore, bootstrap results in this work apply directly to infinite lattices, while Monte Carlo results are estimates obtained from finite-size lattices. Such rigorous results on infinite lattices from bootstrap methods are not obtainable from other methods based on finite-size truncation such as Monte Carlo.
In terms of precision, whether bootstrap results are tighter than Monte Carlo results depends on specifics of the problem. For the examples considered in this work, bootstrap bounds were stronger than Monte Carlo results in some examples (e.g. (1.28)) while they were weaker in other examples (e.g. (1.26)).
To the best of my knowledge, the class of examples considered in this work (stochastic processes of spin degrees of freedom on lattices) can all be studied using previous methods (but on finite lattices) such as Monte Carlo. In contrast, there are other class of physical systems where methods other than bootstrap are not applicable (at least in any obvious way), such as large N matrix quantum mechanics in the 't Hooft limit. My recent works "Thermal Bootstrap of Matrix Quantum Mechanics" and "Nonequilibrium Phase Transitions in Large N Matrix Quantum Mechanics" both investigate such examples. But given that these examples are somewhat far from the systems considered in the current work under consideration, I did not expand on such discussions. Do you recommend me adding to the Future Prospects section brief comments on these examples?
Thank you very much.

---

## Editorial Decision

published